# Hall viscosity and conductivity
# of two-dimensional chiral superconductors

**Félix Rose[1,2,4*], Omri Golan[3] and Sergej Moroz[1,4]**

**1** Physik-Department, Technische Universität München, 85748 Garching, Germany
**2** Max-Planck-Institut für Quantenoptik, 85748 Garching, Germany
**3** Department of Condensed Matter Physics,
Weizmann Institute of Science, Rehovot 76100, Israel
**4** Munich Center for Quantum Science and Technology (MCQST),
80799 München, Germany

* felix.rose@m4x.org

## Abstract

We compute the Hall viscosity and conductivity of non-relativistic two-dimensional chiral superconductors, where fermions pair due to a short-range attractive potential, e.g. $p + \mathrm{i}p$ pairing, *and* interact via a long-range repulsive Coulomb force. For a logarithmic Coulomb potential, the Hall viscosity tensor contains a contribution that is singular at low momentum, which encodes corrections to pressure induced by an external shear strain. Due to this contribution, the Hall viscosity cannot be extracted from the Hall conductivity in spite of Galilean symmetry. For mixed-dimensional chiral superconductors, where the Coulomb potential decays as inverse distance, we find an intermediate behavior between intrinsic two-dimensional superconductors and superfluids. These results are obtained by means of both effective and microscopic field theory.

Received 16 04 -2020
Accepted 18-06- 2020
Published 14-07-2020



# 1 Introduction

Two-dimensional chiral pairing, with its fully gapped Fermi surface and Cooper pairs that coherently carry a finite angular momentum, is a theoretical paradigm of a quantum topological phase of matter [1–3] which is nowadays under intense experimental investigation. It was discovered recently that the chiral A-phase of superfluid $^3$He becomes stable at zero temperature under nanoscale confinement [4,5]. New experimental signatures of topological superconductivity [6] were also reported in thin superconducting films [7,8]. A recent experiment with the 5/2 quantum Hall state [9], which theoretically is believed to be some chirally paired superconductor of composite fermions, reignited the long-term debate about the nature of topological order of this state.

Fermionic chiral paired states exhibit non-dissipative Hall responses because the chiral order parameter breaks time-reversal $T$ and parity $P$ symmetries spontaneously. The well-known Hall conductivity tensor $\sigma_{\mathrm{H}}^{ij}(\omega, \mathbf{q})$ quantifies the response of the U(1) current $\mathbf{J}(\omega, \mathbf{q})$

to a monochromatic electric field $\mathbf{E}(\omega, \mathbf{q})$. The Hall conductivity in a two-dimensional neutral chiral superfluid at zero temperature was computed in [10–14]. In addition to the Hall conductivity, a clean two-dimensional system is characterized by a supplementary non-dissipative Hall response, the Hall (or odd) viscosity tensor $\eta_{\mathrm{o}}^{ijkl}(\omega, \mathbf{q})$, that fixes the (odd under time-reversal) response of the stress tensor to the strain rate [15, 16], see [17] for a review. Recently, observable signatures of the Hall viscosity have been vigorously studied in classical and quantum fluids both theoretically [18–31] and experimentally [32, 33]. In a two-dimensional isotropic system that is invariant under the combined $PT$ symmetry the odd viscosity tensor reduces to two independent components [34], in this paper to be denoted $\eta_{\mathrm{o}}^{(1)}(\omega, \mathbf{q}^2)$ and $\eta_{\mathrm{o}}^{(2)}(\omega, \mathbf{q}^2)$, respectively. If the Hall viscosity tensor is regular in the limit $\mathbf{q} = 0$, only the component $\eta_{\mathrm{o}}^{(1)}$ survives as $\mathbf{q} \to 0$ [15, 16]. For gapped quantum fluids the ratio of the Hall viscosity to the particle number density $n_0$ was argued to be quantized in the units of $\hbar$ as follows [35, 36]

$$\frac{\eta_{\mathrm{o}}^{(1)}}{n_0} = \frac{1}{2}s, \tag{1.1}$$

where $s$ is a rational number that is equal to the average angular momentum per particle. In a neutral $l$-wave chiral superfluid, though gapless, the same relation holds. Here $s = \pm l/2$ with the sign fixed by the chirality of the condensate. As a result, at $\mathbf{q} = 0$ the Hall viscosity coefficient $\eta_{\mathrm{o}}^{(1)}$ does not depend on the topology of the fermionic ground state and thus cannot be used as a diagnostics of topological superconductivity that is characterized by protected chiral Majorana edge modes. It was shown however in [34] that the $\mathbf{q}^2$ dependence of the Hall viscosity tensor contains information about the chiral central charge of the boundary theory, which is determined by the topology of the fermionic ground state.

In single-component Galilean-invariant fluids and solids with a particle number symmetry, the momentum density is proportional to the particle number current, resulting in a Ward identity that ties together the viscosity and conductivity tensors [37–40]. For the two-dimensional chiral superfluid the relation acquires a simple form in the uniform limit $q = |\mathbf{q}| \to 0$

$$\eta_{\mathrm{o}}^{(1)}(\omega) = -\frac{m^2\omega^2}{2}\partial_q^2\sigma_{\mathrm{H}}(\omega, \mathbf{q})\big|_{\mathbf{q}=0}. \tag{1.2}$$

As a result, in this system one can extract the AC Hall viscosity $\eta_{\mathrm{o}}^{(1)}(\omega)$ only from the knowledge of the Hall conductivity at small momentum.

The main aim of this paper is to compute the Hall conductivity and viscosity of a two-dimensional chiral superconductor, where chirally paired fermions couple to a fluctuating electromagnetic field. Two different types of such superconductors can be considered. In a mixed-dimensional superconductor, fermions are restricted to a surface, but the electromagnetic field extends in full three-dimensional space. In such a superconductor plasma oscillations are gapless and charges and vortices exhibit long-range interactions [41]. Electromagnetic response in the mixed-dimensional chiral superconductor was computed in [13].[1] Alternatively, one may consider an intrinsic two-dimensional chiral superconductor, where akin to fermions the electromagnetic field is restricted to the surface. As a result, plasma oscillations are gapped. While this case might seem somewhat contrived, there are two physical motivations to investigate it: (i) In quantum Hall fluids composite fermions couple to a fluctuating emergent $u(1)$ gauge field that is defined in $(2+1)$-dimensional spacetime [43, 44]. As a result, chiral paired states of composite fermions are intrinsic two-dimensional chiral superconductors. (ii) Sufficiently small planar Josephson junction arrays can realize an intrinsic two-dimensional

---

[1]It was also argued recently in [42] that the order parameter and Coulomb fluctuations do not contribute to the Meissner effect in the chiral $p$-wave superconductor.

conventional superconductor [45–48]. This naturally suggests that a planar Josephson junction array of chiral superconducting islands can give rise to an intrinsic two-dimensional chiral superconductor.

In this work we consider non-relativistic fermions, and accordingly, approximate the interaction mediated by the gauge field by an instantaneous Coulomb potential. As a result the problem we study is Galilean invariant. Within this setting we develop a unified theoretical framework for intrinsic and mixed-dimensional chiral superconductors, which we use to extract electromagnetic and gravitational linear responses and investigate the conductivity-viscosity relations following from Galilean symmetry.

Our main results are summarized in Section 2. The rest of the paper is structured as follows: In Section 3 we present a general discussion of electromagnetic and geometric linear responses and the Ward identities which tie them together in Galilean-invariant systems. In Section 4 we develop a low-energy effective field theory of different types of chiral superconductors and present a streamlined calculation of the Hall conductivity and Hall viscosities. In Section 5 we reproduce our results for the Hall responses directly from a canonical microscopic fermionic model of a two-dimensional chiral superconductor. We conclude our work with Section 6, where we provide an outlook for future research.

## 2 Main results and physical picture

In this section, we summarize and explain our main results for the intrinsic two-dimensional chiral superconductor. The derivation of these results and their extension to an arbitrary long-range Coulomb interaction can be found in the following sections.

### 2.1 Setup

Before presenting our results, we emphasize that the Hall conductivity and viscosity computed in this paper are defined as linear responses to *external sources*, as opposed to total fields. As is well known, the total electric field $\mathbf{E} - \nabla\chi$ contains the externally applied field $\mathbf{E}$ and the internal contribution $-\nabla\chi$, where $\chi$ is the electric potential generated by charges in the system. The conductivity we compute is then defined as the response of current to an applied $\mathbf{E}$, and accounts, in particular, for the potential $\chi$ induced due to the applied electric field. This conductivity is physically relevant when an external field $\mathbf{E}$ is applied in the bulk of the system, away from boundaries.

Less appreciated is the analogous decomposition of the strain-rate in the context of the viscosity calculation. The total strain-rate tensor $\dot{u}_{ij} + \partial_{(i}v_{j)}$ contains the externally applied strain $u_{ij}$, which corresponds to a background spatial metric $g_{ij} = \delta_{ij} + 2u_{ij}$, as well as the internal strain-rate $\partial_{(i}v_{j)}$, where $v_j$ is the velocity of particles in the system and the parenthesis denote symmetrization [14]. The viscosity we compute in this paper is defined as the response of stress to an applied $\dot{u}_{ij}$, and accounts, in particular, for the velocity $v_i$ induced due to the applied external strain $u_{ij}$. This viscosity is physically relevant when an external strain rate $\dot{u}_{ij}$ is applied in the bulk of the superconductor away from boundaries, as in Fig. 1. This should be contrasted with standard hydrodynamic scenarios, where $u_{ij} = 0$ and the system is perturbed via boundary conditions [49].

In a crystalline superconductor the strain $u_{ij}$ naturally describes the crystal structure of the ion background [50], and the odd viscosity we compute modifies the dispersion of the corresponding phonon excitations, à la Ref. [51]. It will be useful to analyze our results in terms of the ion displacement field $\xi(\mathbf{x}) = \delta\mathbf{x}$, assumed to be in-plane for simplicity. Then $u_{ij} = \partial_{(i}\xi_{j)}$, and the total strain rate is given by $\partial_{(i}\dot{\xi}_{j)} + \partial_{(i}v_{j)}$, where the ion velocity $\dot{\xi}_i$ and electron velocity $v_i$ enter symmetrically.

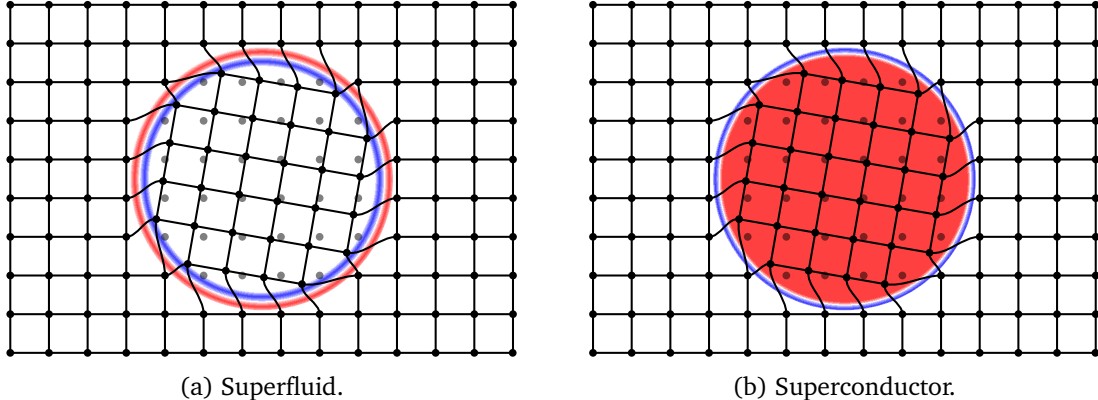

(a) Superfluid.  (b) Superconductor.

Figure 1: Pressure profile induced through the Hall viscosity coefficient $\eta_{\mathrm{o}}^{(2)}$ by an applied external strain in a chiral superfluid (left) and superconductor (right). A disk of radius $R$ is rotated at a frequency $\omega$ around its center, the rotation corresponding to a radial displacement field $\xi_i = \Phi(t,r)\epsilon_{ij}x^j$ with $\Phi(t,r)$ independent of $r$ inside the disk and vanishing outside. The displacement field at a given time is represented by its effect on a black grid, with the untransformed points plotted in grey. The corresponding vorticity $\Omega = -2\dot{\Phi} - r\partial_r\dot{\Phi}$ induces through Eq. (2.4) a change in pressure $\delta p$ shown in color, with red, blue and white corresponding to $\delta p$ positive, negative and vanishing, respectively. In the superfluid case $\eta_{\mathrm{o}}^{(2)}$ is independent of $\mathbf{q}$ at long wavelengths $c_s|\mathbf{q}| \ll \omega$ [see Eq. (4.4)] and the pressure shift $\delta p \sim \mathbf{\nabla}^2\Omega$ is localized at the edge of the disk. On the other hand, for a superconductor, where $\eta_{\mathrm{o}}^{(2)} \sim 1/\mathbf{q}^2$ for $c_s|\mathbf{q}|, \omega \ll \omega_p$ [see Eq. (2.3)], the vorticity inside of the disk is directly related to the pressure shift, $\delta p \sim \Omega$. For a given vorticity, the sign of the resulting pressure change is determined by the chirality of the superfluid/superconductor.

## 2.2 Hall conductivity and viscosity

The Hall conductivity at finite frequency $\omega$ and momentum $\mathbf{q}$ that we find is given by

$$\sigma_{\mathrm{H}}(\omega, \mathbf{q}^2) = \frac{1}{2}\epsilon_{ij}\sigma^{ij}(\omega, \mathbf{q}) = \frac{sn_0}{2m^2}\frac{-\mathbf{q}^2}{\omega^2 - \omega_p^2 - c_s^2\mathbf{q}^2}, \tag{2.1}$$

where we introduced the plasma frequency $\omega_p = \sqrt{e^2 n_0/m}$. In contrast to the neutral chiral superfluid, where $e = 0$, the energy gap of the plasmon excitation ensures that the Hall conductivity always vanishes as $\mathbf{q}^2$ in the limit $\omega, \mathbf{q} \to 0$.

The Hall viscosity tensor of an isotropic $PT$-invariant system is given by [34]

$$\eta_{\mathrm{o}}(\omega, \mathbf{q}) = \eta_{\mathrm{o}}^{(1)}(\omega, \mathbf{q}^2)\sigma^{xz} + \eta_{\mathrm{o}}^{(2)}(\omega, \mathbf{q}^2)[(q_x^2 - q_y^2)\sigma^{0x} - 2q_x q_y \sigma^{0z}], \tag{2.2}$$

and is fixed by the two independent coefficients $\eta_{\mathrm{o}}^{(1)}(\omega, \mathbf{q}^2)$ and $\eta_{\mathrm{o}}^{(2)}(\omega, \mathbf{q}^2)$, where $\sigma^{ab} = \sigma^a \otimes \sigma^b - \sigma^b \otimes \sigma^a$, $a, b = 0, x, z$, are anti-symmetrized tensor products of symmetric Pauli matrices, see Section 3. The physical content of Eq. (2.2) is as follows. The external strain $u_{ij} = \partial_{(i}\xi_{j)}$ can be decomposed into its trace, the compression $u = u_i^i = \mathbf{\nabla}\cdot\xi$, and two traceless shears. The viscous stress $T^{ij} = -\eta_{\mathrm{o}}^{ijkl}\dot{u}_{kl}$ is similarly decomposed into the correction to pressure $\delta p = T_i^i/2$, and two shear stresses. While the usual odd viscosity $\eta_{\mathrm{o}}^{(1)}$ corresponds to a Hall response of the shear stress to a shearing rate, and does not involve the pressure and compression, the component $\eta_{\mathrm{o}}^{(2)}$ induces pressure in response to a shearing rate, and the shear stress in response to a compression rate $\dot{u}$.

For the intrinsic chiral superconductor we find

$$\eta_{\mathrm{o}}^{(1)}(\omega, \mathbf{q}^2) = \frac{sn_0}{2}, \qquad \eta_{\mathrm{o}}^{(2)}(\omega, \mathbf{q}^2) = -\frac{sn_0}{2} \frac{1}{\mathbf{q}^2} \frac{\omega_p^2 + c_s^2 \mathbf{q}^2}{\omega^2 - \omega_p^2 - c_s^2 \mathbf{q}^2}. \tag{2.3}$$

While the first component $\eta_{\mathrm{o}}^{(1)}$ coincides with the result found for the neutral chiral superfluid, the second component $\eta_{\mathrm{o}}^{(2)}$ does not, and exhibits a peculiar $1/\mathbf{q}^2$ singularity at low momenta. Combining Eqs. (2.2) and (2.3), we see that due to this singularity, the contribution of $\eta_{\mathrm{o}}^{(2)}$ to the odd viscosity tensor $\eta_{\mathrm{o}}$ does not vanish in the uniform limit $\mathbf{q} \to 0$, as one may naively expect from Eq. (2.2). In fact, the Hall viscosity tensor of the intrinsic two-dimensional superconductor is ill defined at $\mathbf{q} = 0$, though the superconductor is fully gapped! A similar behavior occurs in the neutral chiral superfluid *only* when it is incompressible, and sound waves propagate with an infinite speed, $c_s = \infty$.[2]

We currently understand the singularity in $\eta_{\mathrm{o}}^{(2)}$ as originating from the instantaneous long-range nature of the Coulomb interaction. In Section 6 we provide an outlook for a future study that will test this understanding.

In order to demonstrate the physical significance of $\eta_{\mathrm{o}}^{(2)}$ and its $1/\mathbf{q}^2$ singularity, we note that Eq. (2.2) implies

$$\delta p = \eta_{\mathrm{o}}^{(2)} \boldsymbol{\nabla}^2 \Omega, \tag{2.4}$$

where $\delta p$ is the pressure relative to the ground state pressure, and $\Omega = \epsilon^{ij} \partial_i \dot{\xi}_j$ is the *applied* vorticity, to be distinguished from the electron vorticity $\epsilon^{ij} \partial_i v_j$, and $\eta_{\mathrm{o}}^{(2)}$ is the operator obtained by Fourier transforming $\eta_{\mathrm{o}}^{(2)}(\omega, \mathbf{q}^2)$ in Eq. (2.3). We see that $\eta_{\mathrm{o}}^{(2)}$ encodes an exotic dissipationless response of pressure to applied vorticity. As follows from Eq. (2.4), a region of space rotating uniformly, where $\Omega$ is constant, does not support pressure variations, unless $\eta_{\mathrm{o}}^{(2)} \sim 1/\mathbf{q}^2$ as $\mathbf{q} \to 0$, which is illustrated in Fig. 1.

A pressure in response to uniform vorticity is currently believed to appear only in the presence of angular momentum non-conservation, as recently discussed in Ref. [60]. In contrast, chiral superconductors and superfluids, studied here, break rotation symmetry only *spontaneously*, and are therefore isotropic and angular momentum conserving.[3] Nevertheless, we see that, due to the $1/\mathbf{q}^2$ singularity in $\eta_{\mathrm{o}}^{(2)}$, chiral superconductors mimic the response $\delta p \sim \Omega$ of angular momentum non-conserving fluids, as shown in Fig. 1b.

In addition to the relation (2.4), Eq. (2.2) also implies

$$\boldsymbol{\nabla} \times \mathbf{f} = \eta_{\mathrm{o}}^{(2)} \boldsymbol{\nabla}^4 \dot{u}, \tag{2.5}$$

where $f^i = -\partial_j T_i^{j}$ is the force density exerted on a test particle by the electron fluid,[4] and $\boldsymbol{\nabla} \times \mathbf{f} = \epsilon^{ij} \partial_i f_j$ is its curl. We see that circulating forces are generated in response to an applied, space-dependent, compression rate. The $1/\mathbf{q}^2$ singularity in $\eta_{\mathrm{o}}^{(2)}$ leads to a reduction $\boldsymbol{\nabla}^4 \to \boldsymbol{\nabla}^2$ in Eq. (2.5).

Eq. (2.5) describes a fully geometric chiral analog of the London diamagnetic response, $\boldsymbol{\nabla} \times \mathbf{J} = -\rho_{\mathrm{L}} \boldsymbol{\nabla}^2 B$, that leads to the Meisner effect in superconductors, where $\rho_{\mathrm{L}} \sim 1/\mathbf{q}^2$ as

---

[2]Naively, a similar $1/\mathbf{q}^2$ singularity appears in the compressible neutral chiral superfluid at $\omega = 0$. However, the viscosity is a response to strain-rate, and is only meaningful for $\omega \neq 0$.

[3]Technically, Eq. (2.4) stems from the momentum dependence of the tensor multiplying $\eta_{\mathrm{o}}^{(2)}$ in Eq. (2.2), while in angular momentum non-conserving fluids, a pressure in response to vorticity derives from the asymmetry of $\eta_{\mathrm{o}}^{ij,kl}$ under $k \leftrightarrow l$.

[4]The force is defined as the time derivative of the momentum density, $f_i = \dot{P}_i$. The momentum conservation equation $\dot{P}_i = -\partial_j T_i^{j}$ holds only in flat space and in the absence of external forces. Here we apply a space-dependent compression, which modifies it to $\dot{P}_i = -\partial_j T_i^{j} + p_0 \partial_i u$, where $p_0$ is the ground state pressure. The momentum source $\partial_i u$ is longitudinal, and does not contribute to $\epsilon^{ij} \partial_i f_j$.

$\mathbf{q} \to 0$. In fact, in *chiral* superconductors, this generalizes to $\nabla \times \mathbf{J} = -\rho_L \nabla^2[B + (s/2)R]$, where $R = \partial_i \partial_j u_{ij} - \nabla^2 u$ is the (linearized) curvature, leading to spontaneous magnetization on curved surfaces [52]. Since, in a Galilean invariant system, the force is related to the electron current by $\mathbf{f} = m\dot{\mathbf{J}}$, Eq. (2.5) implies an additional contribution

$$\nabla \times \mathbf{J} = (\eta_o^{(2)}/m)\nabla^4 u \tag{2.6}$$

due to $\eta_o^{(2)}$, which should be taken into account in future studies of geometrically induced magnetization.

### 2.3 Galilean viscosity-conductivity relation

Another consequence of the $1/\mathbf{q}^2$ singularity in Eq. (2.3) is observed in the relation between the Hall conductivity and the Hall viscosities. Since the theory of a two-dimensional superconductor, where electromagnetic interactions are approximated by an instantaneous Coulomb potential, is Galilean invariant, the Hall conductivity and viscosities are related via the Ward identity

$$m^2 \omega^2 \sigma_H(\omega, \mathbf{q}^2) = -\mathbf{q}^2[\eta_o^{(1)}(\omega, \mathbf{q}^2) - \mathbf{q}^2 \eta_o^{(2)}(\omega, \mathbf{q}^2)], \tag{2.7}$$

see Section 3. Due to the $1/\mathbf{q}^2$ singularity of $\eta_o^{(2)}$, the relation (1.2) for the uniform Hall viscosity $\eta_o^{(1)}(\omega) = \eta_o^{(1)}(\omega, \mathbf{q} = 0)$ does not hold, and the Hall viscosity tensor $\eta_o(\omega, \mathbf{q})$ cannot be extracted from the Hall conductivity alone, even in the uniform limit $\mathbf{q} \to 0$.

Our results for the intrinsic two-dimensional superconductor demonstrate that the two independent components of the Hall viscosity, and $\eta_o^{(1)}$ in particular, cannot generally be extracted from the Hall conductivity. This does not contradict, but should be contrasted, with recent theoretical and experimental work extracting $\eta_o^{(1)}$ from the Hall conductivity or from current profiles [18–22, 33, 38].

As opposed to the intrinsic two-dimensional chiral superconductor, we find that the Ward identity still takes the simple form (1.2) in the mixed-dimensional chiral superconductor.

## 3 Symmetries and transport coefficients

Consider a quantum field theory at zero temperature in two spatial dimensions. The response of the system coupled to an external U(1) gauge field $A_\mu$ and a spatial metric $g_{ij}$ is encoded into the induced action $\mathcal{W}[A_\mu, g_{ij}]$. The induced action can be obtained from the microscopic action $S$ by integrating out the fluctuating degrees of freedom,

$$e^{i\mathcal{W}[A,g]} = \int \mathcal{D}[\cdots] e^{iS[\cdots;A,g]}, \tag{3.1}$$

where the dots stand for the fluctuating fields.

In this Section, we focus on the symmetry properties of the induced action and thus do not specify the precise form of the action $S$, nor the dynamic degrees of freedom we consider, which depend on the scale at which we seek to describe the system. Following [17, 39, 40, 53] we present results in real time[5] and $x = (t, \mathbf{x})$ stands for a $(2+1)$-dimensional space-time variable with Fourier transform $q = (\omega, \mathbf{q})$. We use $\int_x$ as a shorthand for $\int dt \, d^2\mathbf{x}$.

---

[5]However, getting a proper derivation of some expressions, e.g. Eqs. (3.7) and (3.8) or Eq. (3.19) is better done using the imaginary time formalism.

## 3.1 Current, stress tensor and linear response from induced action

In this paper the U(1) gauge field $A_\mu$ and the metric $g_{ij}$ are not dynamical fields but rather act as external sources. By differentiating the microscopic action with respect to the sources, one gets the current densities

$$J^\mu(x) = -\frac{1}{\sqrt{g(x)}}\frac{\delta S}{\delta A_\mu(x)}, \qquad T^{ij}(x) = \frac{2}{\sqrt{g(x)}}\frac{\delta S}{\delta g_{ij}(x)}, \qquad (3.2)$$

with $g = \det(g_{ij})$. The expectation values of the operators $J^\mu$ and $T^{ij}$ can be obtained by replacing the action $S$ in Eq. (3.3) with the induced action $\mathcal{W}$, i.e.,

$$\langle J^\mu(x)\rangle = -\frac{1}{\sqrt{g(x)}}\frac{\delta \mathcal{W}}{\delta A_\mu(x)}, \qquad \langle T^{ij}(x)\rangle = \frac{2}{\sqrt{g(x)}}\frac{\delta \mathcal{W}}{\delta g_{ij}(x)}. \qquad (3.3)$$

The transport coefficients of the theory relate the response of the expectation values of the current densities to an infinitesimal change of the sources. Introducing $h_{ij} = g_{ij} - \delta_{ij}$, one has to leading order in the sources

$$\delta\langle J^i(x)\rangle = \int_{x'} \sigma^{ij}(x-x')[\partial_t A_j(x') - \partial_j A_t(x')] + \mathcal{O}(A^2, h), \qquad (3.4)$$

$$\delta\langle T^{ij}(x)\rangle = -\frac{1}{2}\int_{x'}\lambda^{ijkl}(x-x')h_{kl}(x') - \frac{1}{2}\int_{x'}\eta^{ijkl}(x-x')\partial_t h_{kl}(x') + \mathcal{O}(A, h^2), \qquad (3.5)$$

which defines the conductivity tensor $\sigma^{ij}(x)$, the elastic modulus tensor $\lambda^{ijkl}(x)$ and the viscosity tensor $\eta^{ijkl}(x)$.

As $\mathcal{W}$ is the generating functional of connected correlation functions of currents, the transport coefficients can be expressed in terms of its functional derivatives

$$\mathcal{W}^{(n,m)\{\mu_a\},\{i_b j_b\}}[\{x_a\},\{y_b\};A,g] = \frac{\delta^{n+m}\mathcal{W}[A,g]}{\delta A_{\mu_1}(x_1)\ldots\delta A_{\mu_n}(x_n)\delta g_{i_1 j_1}(y_1)\ldots\delta g_{i_m j_m}(y_m)}. \qquad (3.6)$$

In the following we denote the vertices evaluated in flat space $h_{ij} = 0$ and vanishing U(1) gauge field $A_\mu$ by $\mathcal{W}^{(n,m)\{\mu_a\},\{i_b j_b\}}(\{x_a\},\{y_b\})$. Due to translation invariance the two-point vertices are diagonal in Fourier space, for instance $\mathcal{W}^{(2,0)ij}(q,q') = \mathcal{W}^{(2,0)ij}(q)\delta_{q,-q'}$.

By further differentiating Eq. (3.3) and using the definitions Eqs. (3.4) and (3.5) one gets

$$\mathcal{W}^{(2,0)ij}(\omega,\mathbf{q}) = i\omega^+\sigma^{ij}(\omega,\mathbf{q}), \qquad (3.7)$$

$$\mathcal{W}^{(0,2)ijkl}(\omega,\mathbf{q}) = \frac{1}{4}\langle T^{ij}\rangle\delta^{kl} - \frac{1}{4}\lambda^{ijkl}(\omega,\mathbf{q}) + \frac{i\omega^+}{4}\eta^{ijkl}(\omega,\mathbf{q}), \qquad (3.8)$$

with $\langle T^{ij}\rangle = \langle T^{ij}(\omega = 0, \mathbf{q} = 0)\rangle$ being the homogeneous expectation value of the stress tensor for vanishing sources. In Eqs. (3.7) and (3.8), the left-hand-sides stand for the retarded correlation functions and $\omega^+ = \omega + i0^+$. The infinitesimal imaginary part $0^+$ enforces causality in Eqs. (3.4) and (3.5).

## 3.2 Tensor decompositions of conductivity and viscosity

Both conductivity and viscosity must transform as tensors under SO(2) rotations. The conductivity of an isotropic system can be decomposed as[6]

$$\sigma^{ij}(\omega,\mathbf{q}) = (q^i q^j/\mathbf{q}^2)\sigma_{\text{L}}(\omega,\mathbf{q}^2) + (\delta^{ij} - q^i q^j/\mathbf{q}^2)\sigma_{\text{T}}(\omega,\mathbf{q}^2) + \epsilon^{ij}\sigma_{\text{H}}(\omega,\mathbf{q}^2), \qquad (3.9)$$

---

[6]We exclude in Eq. (3.9) an SO(2) invariant term $\propto q^i\epsilon^{jk}q_k + \epsilon^{ik}q_k q^j$. Such a term leads to dissipation $J^i E_i$ with an unconstrained sign, which is in conflict with the second law of thermodynamics.

with $\epsilon^{xy} = +1$, $\epsilon^{ij} = -\epsilon^{ji}$ the Levi-Civita symbol, and $\sigma_{\rm L}$, $\sigma_{\rm T}$ and $\sigma_{\rm H}$ being the longitudinal, transverse and Hall conductivities, respectively. The longitudinal and transverse components constitute the symmetric part of the conductivity tensor, satisfying $\sigma^{ij} = \sigma^{ji}$, while the Hall component fixes its antisymmetric part, $\sigma^{ij} = -\sigma^{ji}$, and describes dissipationless transport of particles. The Hall conductivity $\sigma_{\rm H}(\omega, \mathbf{q})$ vanishes unless time-reversal symmetry is broken.

A similar decomposition is possible for the viscosity tensor. First, we note that by construction, it must be invariant under exchange of either the first or second pair of indices, $\eta^{ijkl} = \eta^{jikl} = \eta^{ijlk}$, since the metric is symmetric, $g_{ij} = g_{ji}$. It can be written as a sum of the even and odd tensors $\eta_{\rm e}$ and $\eta_{\rm o}$ satisfying

$$\eta_{\rm e}^{ijkl} = \eta_{\rm e}^{klij}, \qquad\qquad \eta_{\rm o}^{ijkl} = -\eta_{\rm o}^{klij}. \qquad\qquad (3.10)$$

The symmetric part includes shear and bulk viscosities, as well as, at finite $\mathbf{q}$, other even tensors which can be constructed using additionally the momentum $q_i$. In this paper we restrict our attention to the antisymmetric part of the viscosity tensor, known as Hall or odd viscosity [15, 16]. Much like for the conductivity, the dissipationless[7] odd viscosity is a signature of the breaking of time-reversal symmetry. It has been shown in [34] that, in an isotropic system that is symmetric under the combination of parity and time-reversal symmetries ($PT$ symmetry) the Hall viscosity tensor is fixed by only two independent components $\eta_{\rm o}^{(1)}$ and $\eta_{\rm o}^{(2)}$[8]

$$\eta_{\rm o}(\omega, \mathbf{q}) = \eta_{\rm o}^{(1)}(\omega, \mathbf{q}^2)\sigma^{xz} + \eta_{\rm o}^{(2)}(\omega, \mathbf{q}^2)[(q_x^2 - q_y^2)\sigma^{0x} - 2q_x q_y \sigma^{0z}], \qquad (3.11)$$

where the $\sigma^{ab}$ matrices are antisymmetrized tensor products of the Pauli matrices $\sigma^a$

$$(\sigma^{ab})^{ijkl} = (\sigma^a)^{ij}(\sigma^b)^{kl} - (\sigma^b)^{ij}(\sigma^a)^{kl}. \qquad (3.12)$$

Since the momentum-dependent tensor that multiplies $\eta_{\rm o}^{(2)}$ in Eq. (3.11) vanishes at $\mathbf{q} = 0$, one expects the response to the homogenous ($\mathbf{q} = 0$) perturbation to be fully determined by the first term proportional to $\eta_{\rm o}^{(1)}$ [16], often identified in the literature with the Hall viscosity [15, 35, 39]. However, as we discuss later in Sections 4 and 5, in some cases the coefficient $\eta_{\rm o}^{(2)}$ is singular like $1/\mathbf{q}^2$ in the $\mathbf{q} \to 0$ limit, such that $\eta_{\rm o}^{ijkl}(\omega, \mathbf{q} = 0) \neq \lim_{\mathbf{q} \to 0} \eta_{\rm o}^{ijkl}(\omega, \mathbf{q})$. Indeed, when this happens the second term in Eq. (3.11) doesn't vanish at small but finite $\mathbf{q}$ and cannot be dropped carelessly, while the viscosity in the homogenous limit is determined by the single coefficient $\eta_{\rm o}^{(1)}$.

### 3.3  Galilean Ward identities

In a Galilean-invariant system composed of a single species of particles, the conductivity and viscosity are not independent, as the transport of electrical charge is tied to the transport of momentum density. The formal expression of this statement comes from the Ward identities relating the correlation fuctions of the current $J^i$ and the stress tensor $T^{ij}$ to each other [37, 39, 40].

Consider a non-relativistic theory whose action $S$ is invariant under global U(1) transformations as well as under global spatial translations. By coupling to an external U(1) gauge

---

[7]Dissipation can arise from $\sigma_{\rm H}$ and $\eta_{\rm o}$, if these are not even functions of $\omega$. However, this does not occur for systems at equilibrium, like the chiral superfluids and superconductors we consider. Indeed, the relations (3.7) and (3.8) together with the fact that the functional derivatives in Eq. (3.6) can be taken in any order imply $\sigma^{ij}(\omega) = -\sigma^{ji}(-\omega)$ and $\eta^{ijkl}(\omega) = -\eta^{klij}(-\omega)$.

[8]Our definition of the viscosity tensor $\eta$ in Eq. (3.5) follows the standard hydrodynamic convention. It agrees with Refs. [14, 39] and is opposite to that used in [34]. Our definition Eq. (3.11) of the viscosity coefficients from $\eta$ follows [34]. In particular, comparison with Refs. [14, 39] is obtained via $\eta^H = \eta_H = -\eta_{\rm o}^{(1)}$.

field $A_\mu$ and defining the theory in space with a spatial metric $g_{ij}$, we can promote these two global symmetries to local gauge invariance, provided $A_\mu$ and $g_{ij}$ transform as [53]

$$\delta A_t = -\partial_t \alpha - \xi^k \partial_k A_t - A_k \partial_t \xi^k, \tag{3.13}$$

$$\delta A_i = -\partial_i \alpha - \xi^k \partial_k A_i - A_k \partial_i \xi^k + m g_{ik} \partial_t \xi^k, \tag{3.14}$$

$$\delta g_{ij} = -\xi^k \partial_k g_{ij} - g_{ik} \partial_j \xi^k - g_{kj} \partial_i \xi^k \tag{3.15}$$

under infinitesimal U(1) gauge transformations and spatial diffeomorphisms with respective parameters $\alpha(x)$ and $\xi^i(x)$. The induced action inherits the symmetries of the original action and as a result,

$$\mathcal{W}[A_\mu + \delta A_\mu, g_{ij} + \delta g_{ij}] = \mathcal{W}[A_\mu, g_{ij}]. \tag{3.16}$$

As the invariance is valid for any infinitesimal transform, Eq. (3.16) implies two independent identities that holds at any point in spacetime and any value of the sources. Expressed in terms of expectation values, these read

$$\frac{1}{\sqrt{g}} \partial_t (\sqrt{g} \langle J^t \rangle) + \nabla_i \langle J^i \rangle = 0, \tag{3.17}$$

$$\frac{1}{\sqrt{g}} m \partial_\tau (\sqrt{g} \langle J_k \rangle) + \nabla_i \langle T_k^i \rangle = E_k \langle J^t \rangle + \varepsilon_{ik} \langle J^i \rangle B. \tag{3.18}$$

Eqs. (3.17) and (3.18) are respectively the continuity equations for the U(1) current and momentum density in the background of a general U(1) gauge field $A_\mu$ and the metric $g_{ij}$. Here we introduced the covariant Levi-Civita derivative $\nabla_i$, the Levi-Civita tensor $\varepsilon_{ij} = \sqrt{g} \epsilon_{ij}$, $\varepsilon^{ij} = (1/\sqrt{g}) \epsilon^{ij}$, the electric field $E_j = \partial_t A_j - \partial_j A_t$ and the magnetic field $B = \varepsilon^{ij} \partial_i A_j$.

Since Eqs. (3.17) and (3.18) are valid for any configurations of the sources, it is possible to take further derivatives to obtain relations between $n$-point correlation functions. For the two-point functions we derive these in Appendix A. As a result, we find a relation between the transport coefficients[9]

$$m^2 (\omega^+)^2 \sigma^{ij}(\omega, \mathbf{q}) = q_k q_l \eta^{ikjl}(\omega, \mathbf{q}) - \frac{1}{i\omega^+} q^i q^j \kappa^{-1}, \tag{3.19}$$

where $\kappa^{-1} = -V (\partial P / \partial V)_{S,N}$ is the inverse compressibility. Projecting it on the antisymmetric part gives

$$m^2 (\omega^+)^2 \sigma_H(\omega, \mathbf{q}^2) = -\mathbf{q}^2 [\eta_o^{(1)}(\omega, \mathbf{q}^2) - \mathbf{q}^2 \eta_o^{(2)}(\omega, \mathbf{q}^2)], \tag{3.20}$$

which is valid for all $\mathbf{q}$, $\omega$ provided $\lambda^{ijkl}(\omega, \mathbf{q})$ has no odd part.

# 4 Effective field theory

At low energies and long wave-lengths the collective degrees of freedom of a nonrelativistic superfluid are determined by spontaneous symmetry breaking and the Galilean-invariant dynamics can be encoded in a non-linear effective action of Goldstone bosons [53, 54]. The chiral ground state of two-dimensional fermions paired in the $l^{\text{th}}$ partial wave has the order parameter $\langle \psi_\mathbf{q} \psi_{-\mathbf{q}} \rangle \sim \Delta_\mathbf{q} = (q_x + i q_y)^l |\Delta_0|$ [1, 3] which has a non-trivial phase winding around the Fermi surface.[10] As a result, the global particle number U(1)$_N$ symmetry and the

---

[9]As one can see in Appendix A, to get (3.19) we must replace the elastic tensor $\lambda^{ijkl}(\omega, \mathbf{q})$ with its $\mathbf{q} = 0$, $\omega = 0$ expression; the Ward identity is thus valid for all momenta and frequencies up to finite $\mathbf{q}$ corrections that originate from $\lambda^{ijkl}(\omega, \mathbf{q})$.

[10]Due to the Pauli principle, the chirality parameter $l$ for spinless fermions must be odd, while that of spin-full fermions where $\langle \psi_{\uparrow,\mathbf{q}} \psi_{\downarrow,-\mathbf{q}} \rangle \sim \Delta_\mathbf{q} = (q_x + i q_y)^l |\Delta_0|$ must be even.

rotation $SO(2)_R$ symmetry are both spontaneously broken by the condensate, while a special linear combination $U(1)_D$ of these two symmetries leaves the order parameter invariant. The spontaneous symmetry breaking thus has the form $U(1)_N \times SO(2)_R \to U(1)_D$ implying that there is only one Goldstone mode in the energy spectrum. The effective field theory (EFT) of this Goldstone boson in a Galilean-invariant two-dimensional chiral superfluid was developed in [14, 34, 55].

We first briefly review this theory in Section 4.1 and present the Hall conductivity and Hall viscosity that were extracted from it. Subsequently, we develop in Section 4.2 the effective theory of a two-dimensional intrinsic superconductor by incorporating the effects of the instantaneous Coulomb interaction and extract from the induced action the Hall responses of these chirally paired superconducting states. Finally, in Section 4.3 we extend the calculation to the case of general long-ranged interactions, encompassing the case of mixed-dimensional chiral superconductors. A comprehensive analysis of linear response in chiral superconductors is presented in Appendix C.

## 4.1 Chiral superfluid

To first order in derivatives, the effective action of the Goldstone field $\theta$, coupled to a background spatial metric $g_{ij}$ and $U(1)_N$ gauge field $A_\mu$, is fixed by the thermodynamic pressure as a function of chemical potential $P(\mu)$ by

$$S[\theta; A, g] = \int_x \sqrt{g} P(X), \tag{4.1}$$

where $X = D_t \theta - \frac{g^{ij}}{2m} D_i \theta D_j \theta$ and the covariant derivative $D_\mu \theta = \partial_\mu \theta - A_\mu - s\omega_\mu$ [14, 53]. Here the chirality parameter is $s = \pm l/2$ for a chiral superfluid paired in the $l^{\text{th}}$ partial wave, and the spin connection $\omega_\mu$ is constructed from a pair of orthonormal vielbein vectors[11] $e_i^a$ as

$$\omega_t = \frac{1}{2}\left(\epsilon^{ab} e^{aj} \partial_t e_j^b + B\right), \qquad \omega_i = \frac{1}{2}\epsilon^{ab} e^{aj} \nabla_i e_j^b = \frac{1}{2}\left(\epsilon^{ab} e^{aj} \partial_i e_j^b - \varepsilon^{jk} \partial_j g_{ik}\right). \tag{4.2}$$

The superfluid density is given by $n = -\left(1/\sqrt{g}\right)\delta S/\delta A_t = P'(X)$, where the prime indicates a derivative. In the ground state $\theta = \mu t + \text{const.}$, in which case $X$ reduces to the chemical potential $\mu$ and the density $n$ reduces to the thermodynamic expression $n_0 = P'(\mu)$.

Following a standard linear response calculation, the Hall transport coefficients were extracted from the EFT (4.1) in [14, 34, 55]. The Hall conductivity was found to be equal to

$$\sigma_{\text{H}}(\omega, \mathbf{q}^2) = \frac{sn_0}{2m^2} \frac{-\mathbf{q}^2}{\omega^2 - c_s^2 \mathbf{q}^2}, \tag{4.3}$$

where $c_s = \sqrt{\partial P/\partial n}$ is the speed of sound. The two independent components of the Hall viscosity tensor (3.11) were calculated,[12]

$$\eta_{\text{o}}^{(1)}(\omega, \mathbf{q}^2) = \frac{sn_0}{2}, \quad \eta_{\text{o}}^{(2)}(\omega, \mathbf{q}^2) = -\frac{sn_0}{2} \frac{c_s^2}{\omega^2 - c_s^2 \mathbf{q}^2}. \tag{4.4}$$

The action (4.1) is invariant under local U(1) gauge transformations and spatial diffeomorphisms [14, 34, 55] resulting in the Ward identity (3.20) that relates the Hall conductivity

---

[11]Out of the vielbein pair $e_i^a$ one can construct the spatial metric $g_{ij} = e_i^a e_j^a$ and the Levi-Civita tensor $\varepsilon_{ij} = \epsilon^{ab} e_i^a e_j^b$. Both of these tensors are invariant under local $SO(2)_v$ rotation of vielbeins in internal space labeled by the index $a$. Hence, for a given metric $g_{ij}$ the vielbeins are not uniquely defined, and while we write for clarity the action (4.1) as a functional of the metric, it should rather read $S[\theta; A, e]$.

[12]The relative sign between $\eta_{\text{o}}^{(1)}$ and $\eta_{\text{o}}^{(2)}$ in Eq. (4.4) corrects a typo in Ref. [34].

and viscosity tensors. In the homogeneous limit $\mathbf{q} \to 0$ the Hall viscosity tensor $\eta_o(\omega)$ (3.11) reduces to only one component $\eta_o^{(1)}(\omega)$ and the odd version of the conductivity-viscosity Ward identity takes the simple form (1.2).

It is clear from Eq. (4.4) that the component $\eta_o^{(2)}(\omega, \mathbf{q}^2)$ of the Hall viscosity tensor becomes formally singular at $\mathbf{q} = 0$ in the incompressible limit $c_s \to \infty$. We will see in the following that a similar singularity arises in a chiral two-dimensional intrinsic superconductor.

## 4.2 Intrinsic two-dimensional chiral superconductor

In order to incorporate an instantaneous logarithmic Coulomb interaction between fermions, we couple the superfluid effective theory to a mediating, or Hubbard-Stratonovich, scalar field $\chi$. The effective action of an intrinsic two-dimensional chiral superconductor is then

$$S[\theta, \chi; A, g] = \int_x \sqrt{g} \left[ P(X - \chi) + \bar{n}\chi + \frac{1}{2e^2} g^{ij} \partial_i \chi \partial_j \chi \right]. \tag{4.5}$$

The uniform background density $\bar{n}$ ensures that the overall system is electrically neutral, and $e$ corresponds to the electric charge of a microscopic fermion. The equation of motion for $\chi$ is the two-dimensional Poisson equation

$$\frac{1}{\sqrt{g}} \partial_i \left( g^{ij} \sqrt{g} \partial_j \chi \right) = -e\delta Q, \tag{4.6}$$

where $\delta Q = e[P'(X - \chi) - \bar{n}]$ is the total charge density.

Since the Coulomb potential is instantaneous, it does not break Galilean symmetry. More generally, we impose that the Coulomb field $\chi$ transforms as a scalar under spatial diffeomorphisms and does not transform under local $U(1)_N$ transformations. As a result, the action (4.5) is invariant under both transformations, and the Ward identity (3.20) remains intact.

We now turn to the computation of linear response functions, namely the Hall conductivity and viscosity, based on the action Eq. (4.5). We perform the computation within the random phase approximation (RPA), and to leading order in derivatives. These approximations amount to a quadratic expansion of the action Eq. (4.5) in all fields, and are discussed in Appendix C.1.3. We write the Goldstone field $\theta = \mu t - \varphi$, where the first term represents the ground state contribution, while the second term denotes the fluctuating part of the field. We define the Lagrangian density $\mathcal{L}$ by $S = \int_x \sqrt{g} \mathcal{L}$. Expanding $\mathcal{L}$ to quadratic order in the fluctuations $\varphi$ and the Coulomb field $\chi$, we find

$$\mathcal{L} = P(\mu) - P'(\mu)\left( \chi + D_t \varphi + \frac{g^{ij}}{2m} D_i \varphi D_j \varphi \right)$$
$$+ \frac{1}{2} P''(\mu)\left[ \chi^2 + (D_t \varphi)^2 + 2\chi D_t \varphi \right] + \bar{n}\chi + \frac{1}{2e^2} g^{ij} \partial_i \chi \partial_j \chi, \tag{4.7}$$

where $D_\mu \varphi = \partial_\mu \varphi + A_\mu + s\omega_\mu$ and primes denote derivatives with respect to the chemical potential. Introducing $P_0 = P(\mu)$, $n_0 = P'(\mu)$ and $P''(\mu) = n_0/mc_s^2$, applying the charge neutrality condition $n_0 = \bar{n}$ and rearranging the terms, we get

$$\mathcal{L} = P_0 - n_0 D_t \varphi - \frac{n_0}{2m} g^{ij} D_i \varphi D_j \varphi + \frac{1}{2} \frac{n_0}{mc_s^2} (D_t \varphi)^2 + \frac{n_0}{mc_s^2} \chi D_t \varphi + \frac{1}{2e^2} g^{ij} \partial_i \chi \partial_j \chi + \frac{1}{2} \frac{n_0}{mc_s^2} \chi^2. \tag{4.8}$$

At this stage we would like to compute the induced action $\mathcal{W}[A, g]$ by preforming the Gaussian functional integration over the Coulomb and Goldstone fields $\chi$ and $\varphi$.[13] First, we integrate over the Coulomb field $\chi$ and find

---

[13] Gaussian functional integration in the presence of a background metric $g_{ij}$ is briefly summarized in Appendix B.

$$\mathcal{L} = P_0 - n_0 D_t \varphi - \frac{n_0}{2m} g^{ij} D_i \varphi D_j \varphi$$
$$+ \frac{1}{2} \frac{n_0}{mc_s^2} (D_t \varphi)^2 + \frac{1}{2} \left( \frac{n_0}{mc_s^2} D_t \varphi \right) \frac{1}{\mathbf{\nabla}^2/e^2 - n_0/mc_s^2} \left( \frac{n_0}{mc_s^2} D_t \varphi \right)$$
$$= P_0 - n_0 D_t \varphi - \frac{n_0}{2m} g^{ij} D_i \varphi D_j \varphi + \frac{1}{2} \frac{n_0}{m} D_t \varphi \tilde{c}_s^{-2} D_t \varphi, \tag{4.9}$$

where we introduced the renormalized momentum-dependent speed of sound operator as

$$\tilde{c}_s^2 = c_s^2 - \omega_p^2/\mathbf{\nabla}^2, \tag{4.10}$$

with $\omega_p = \sqrt{e^2 n_0/m}$ the plasma frequency. The Lagrangian (4.9) can alternatively be obtained following the derivation in Appendix C.1. We are now ready to integrate out the Goldstone field $\varphi$. First, we put the functional integral into the standard Gaussian form by performing several integrations by parts. Next, we follow Appendix B and obtain the induced action

$$\mathcal{W}[A,g] = \int_x \sqrt{g} \left\{ P_0 - n_0 \left( A_t + \frac{g^{ij}}{2m} A_i A_j \right) + \frac{1}{2} \frac{n_0}{m} A_t \tilde{c}_s^{-2} A_t \right.$$
$$\left. - \frac{n_0}{2m} \left[ m\dot{f} + \mathbf{\nabla} \cdot \mathsf{A} - \tilde{\partial}_t \tilde{c}_s^{-2} A_t \right] \frac{1}{\mathbf{\nabla}^2 - \tilde{\partial}_t \tilde{c}_s^{-2} \partial_t} \left[ m\dot{f} + \mathbf{\nabla} \cdot \mathsf{A} - \tilde{\partial}_t \tilde{c}_s^{-2} A_t \right] \right\}, \tag{4.11}$$

where we introduced $f = \log \sqrt{g}$, $\tilde{\partial}_t = \partial_t + \dot{f}$ and $\mathsf{A}_\mu = A_\mu + s\omega_\mu$. Expanding now the induced action around flat space and following Appendix C.2, the induced action can be written in the covariant form in Fourier space

$$\mathcal{W}[A,g] = \int_x \left[ 2P_0 h - n_0 A_t \right.$$
$$+ \frac{1}{2} \frac{n_0}{m} \frac{\tilde{c}_s^2 B^2 - E^2 + (is/m) E^i q_i B - (s^2/4m^2) \mathbf{q}^2 B^2}{\omega^2 - \tilde{c}_s \mathbf{q}^2}$$
$$\left. + 2n_0 \frac{\tilde{c}_s^2 h [iq_i E^i + (s/2m) \mathbf{q}^2 B] - m\tilde{c}_s^2 \omega^2 h^2}{\omega^2 - \tilde{c}_s^2 \mathbf{q}^2} \right], \tag{4.12}$$

where $h_{ij} = g_{ij} - \delta_{ij}$, $h = h_i^i$, and $E_i = \partial_t A_i - \partial_i A_t$ and $B = \varepsilon^{ij} \partial_i A_j$ are the electric and magnetic fields constructed with $A_\mu$. As we argue in Appendix C, the induced action of the chiral superconductor is identical to the one found for the chiral superfluid provided the renormalized speed of sound $\tilde{c}_s$ is used.

It is straightforward now to extract electromagnetic and gravitational linear responses from either (4.11) or (4.12). Their comprehensive calculation and analysis based on the latter expression of the induced action is performed in Appendix C.2. Here we present results for the Hall conductivity and viscosities. Using Eq. (3.7) we find the Hall conductivity in flat space and $A_\mu = 0$

$$\sigma_{\mathrm{H}}(\omega, \mathbf{q}^2) = \frac{1}{2} \epsilon_{ij} \sigma^{ij} = \frac{s n_0}{2m^2} \frac{-\mathbf{q}^2}{\omega^2 - \omega_p^2 - c_s^2 \mathbf{q}^2}. \tag{4.13}$$

This result resembles the Hall conductivity of the chiral superfluid (4.3) with the only difference in the denominator stemming from the gapped nature of the collective plasmon mode. As a result, at small frequency $\omega$ and momentum $\mathbf{q}$ the Hall conductivity vanishes as a quadratic function of the momentum

$$\sigma_{\mathrm{H}}(\omega \to 0, \mathbf{q} \to 0) = \frac{s}{2me^2} \mathbf{q}^2. \tag{4.14}$$

In contrast to the Hall conductivity of the gapless superfluid (4.3), this result is unique and does not depend on the order of limits.

Using now Eqs. (3.8) and (3.11) we extract the two independent components of the Hall viscosity tensor

$$\eta_o^{(1)}(\omega, \mathbf{q}^2) = \frac{sn_0}{2}, \quad \eta_o^{(2)}(\omega, \mathbf{q}^2) = -\frac{sn_0}{2}\frac{1}{\mathbf{q}^2}\frac{\omega_p^2 + c_s^2\mathbf{q}^2}{\omega^2 - \omega_p^2 - c_s^2\mathbf{q}^2}. \tag{4.15}$$

We find that the component $\eta_o^{(1)}$ is identical to the one found for the chiral superfluid (4.4). In other words, the instantaneous Coulomb interaction does not affect $\eta_o^{(1)}$. On the other hand, the component $\eta_o^{(2)}$ is modified. Most notably, in the homogeneous limit $\mathbf{q} \to 0$ it diverges as $1/\mathbf{q}^2$. We thus conclude that due to the long-range Coulomb potential the homogeneous limit of the Hall viscosity tensor (3.11) is ill-defined since the result depends on the direction of the vanishing vector $\mathbf{q}$. We attribute this peculiar singularity to the instantaneous nature of the Coulomb potential, and in Section 6 will provide an outlook on its fate in a model which involves photons that propagate with a finite speed of light.

It is straightforward to check that the Hall conductivity and viscosity found above satisfy the Ward identity (3.20) that follows from the Galilean symmetry of the chiral superconductor, where the electromagnetic interaction is approximated by the instantaneous Coulomb interaction. In the regime of small momentum $\mathbf{q}$, in contrast to the chiral superfluid, the contribution of $\eta_o^{(2)}$ to the Ward identity does not drop. The Hall conductivity encodes information about a particular combination of the two independent components of the Hall viscosity tensor, but is not sufficient to fix either of them separately.

## 4.3 Mixed-dimensional chiral superconductor

In a mixed-dimensional superconductor the Coulomb potential decays as $1/|\mathbf{r}|$ at large distances which is dictated by the three-dimensional nature of the electromagnetic field. It is straightforward to generalize the effective theory developed in the previous section to this case. To include a generic power-law decaying interaction, we start from the effective action

$$S[\theta, \chi; A, g] = \int_x \sqrt{g}\left[P(X - \chi) + \bar{n}\chi + \frac{1}{2e^2}\chi(-\boldsymbol{\nabla}^2)^{\alpha/2}\chi\right], \tag{4.16}$$

where $\boldsymbol{\nabla}^2$ is the covariant Laplace operator and $0 \le \alpha \le 2$. In flat space the scalar field $\chi$ mediates an instantaneous repulsive central potential that decays as $|\mathbf{r}|^{\alpha-2}$. The special case $\alpha = 1$ corresponds to the mixed-dimensional superconductor. The case $\alpha = 2$ is the intrinsic superconductor discussed in the previous subsection, where the Coulomb potential is logarithmic.

Following the steps of the previous subsection one arrives at the induced action (4.11), where now the inverse square of the renormalized speed of sound is

$$\tilde{c}_s^{-2} = \frac{(-\boldsymbol{\nabla}^2)^{\alpha/2}}{c_s^2(-\boldsymbol{\nabla}^2)^{\alpha/2} + \omega_p^2}. \tag{4.17}$$

Here as before we introduced $\omega_p^2 = e^2 n_0/m$. We stress that $\omega_p$ is a frequency that defines the plasmon gap only in the case of the intrinsic superconductor, i.e., for $\alpha = 2$. For all $0 < \alpha < 2$ plasmon collective modes are gapless and the units of $\omega_p$ are $[\omega q^{(\alpha-2)/2}]$.

The induced action defined by Eq. (4.11) contains all necessary information to extract electromagnetic and gravitational linear responses. This is discussed in detail for a generic

value of $\alpha$ in Appendix C.2. Here we present the Hall responses. The Hall conductivity is

$$\sigma_{\mathrm{H}}(\omega, \mathbf{q}^2) = \frac{1}{2}\epsilon_{ij}\sigma^{ij} = \frac{sn_0}{2m^2}\frac{-\mathbf{q}^2}{\omega^2 - \omega_p^2|\mathbf{q}|^{2-\alpha} - c_s^2\mathbf{q}^2}. \tag{4.18}$$

In a mixed-dimensional superconductor ($\alpha = 1$) the plasmon mode is gapless and disperses as $\sqrt{|\mathbf{q}|}$ as low momenta. This implies that the limits $\omega \to 0$ and $\mathbf{q} \to 0$ of the Hall conductivity do not commute. The Hall viscosities extracted from the induced action are

$$\eta_{\mathrm{o}}^{(1)}(\omega, \mathbf{q}^2) = \frac{sn_0}{2}, \quad \eta_{\mathrm{o}}^{(2)}(\omega, \mathbf{q}^2) = -\frac{sn_0}{2}\frac{1}{|\mathbf{q}|^\alpha}\frac{\omega_p^2 + c_s^2|\mathbf{q}|^\alpha}{\omega^2 - \omega_p^2|\mathbf{q}|^{2-\alpha} - c_s^2\mathbf{q}^2}. \tag{4.19}$$

The component $\eta_{\mathrm{o}}^{(2)}$ diverges as $|\mathbf{q}|^{-\alpha}$ in the homogeneous $\mathbf{q} \to 0$ limit. We thus conclude that for any $0 \le \alpha < 2$ the $\eta_{\mathrm{o}}^{(2)}$ contribution to the Hall viscosity (3.11) tensor vanishes for $\mathbf{q} \to 0$. As a result, the gapped intrinsic superconductor, where the Hall viscosity tensor is ill-defined in the limit $\mathbf{q} \to 0$, is an exceptional case.

An explicit calculation confirms that the conductivity-viscosity Ward identity (3.20) is satisfied for a generic value of $\alpha$. In the homogeneous case, if $\alpha \ne 2$, this Ward identity simplifies to the form (1.2).

# 5 Microscopic theory

In this Section, starting from a microscopic theory of fermions we derive the transport coefficients of the chiral paired states. First, we consider the case of the neutral chiral superfluid, where spinless fermions attract each other via a short-range potential. Next, we study the superconductor by including into the microscopic model the effects of the long-range repulsive Coulomb interaction.

The starting point of our discussion is the action of spinless fermions $\psi$ interacting via a separable $p$-wave short-range potential in space with arbitrary metric $g_{ij}$ and in presence of a $U(1)_N$ gauge field $A_\mu$,

$$S = \int_x \sqrt{g}\left\{\psi^*\left[\partial_\tau - iA_0 + \frac{g^{ij}(p_i - A_i)(p_j - A_j)}{2m} - \mu\right]\psi - \lambda g^{ij}(\psi^* p_i \psi^*)(\psi p_j \psi)\right\}, \tag{5.1}$$

where $x = (\tau, \mathbf{r})$ is the $(2+1)$-dimensional space-time coordinate, $p_i = -i\nabla_i$ the momentum operator, $m$ the fermion mass and $\lambda$ the interaction strength. Note that the $U(1)_N$ gauge field doesn't appear in the interaction term due to the Pauli principle.

In Section 5.1, we derive from Eq. (5.1) the Bardeen–Cooper–Schrieffer (BCS) theory of $p$-wave pairing. An induced action for the external fields is formally obtained in Section 5.2, from which we compute the transport coefficients in Section 5.3. In Section 5.4, we extend these results in presence of long-range interactions such as the Coulomb interaction.

In this Section the microscopic action Eq. (5.1) is formulated in Euclidian (imaginary) time, $t \to -i\tau$, which is more convenient for the linear response computation. The induced action is given by the functional integral

$$\exp(-\mathcal{W}[A, g]) = \int \mathcal{D}[\psi, \psi^*]\exp(-S[\psi, \psi^*; A, g]) \tag{5.2}$$

and like its real-time counterpart, it is the generating functional of the connected correlation functions. We compute two-point correlation functions $f(i\omega_n, q_x, q_y)$ depending on a Matsubara frequency $i\omega_n$ from which the real-time, retarded dynamical correlation functions $f^{\mathrm{R}}(\omega, \mathbf{q})$

are obtained after analytic continuation $i\omega_n \to \omega^+ = \omega + i0^+$. Notice that upon going to imaginary time, the time component of the $U(1)_N$ gauge field $A_t$ transforms like a time derivative, $A_t \to iA_0$. This should be taken into account when going back to real time to determine the retarded correlation functions.

## 5.1 BCS action

To decouple the $p$-wave interaction, we introduce an auxiliary two-component bosonic (complex) field $\overline{\mathbf{\Delta}}$, and perform a Hubbard-Stratonovitch transform to get

$$S[\psi, \psi^*, \overline{\mathbf{\Delta}}; A, g] = \int_x \sqrt{g} \left\{ \psi^* \left[ \partial_\tau - iA_0 + \frac{g^{ij}(p_i - A_i)(p_j - A_j)}{2m} - \mu \right] \psi \right.$$
$$\left. - \frac{1}{2}(\overline{\Delta}^i)^*(\psi p_i \psi) - \frac{1}{2}\overline{\Delta}^i(\psi^* p_i \psi^*) + \frac{1}{4\lambda}|\overline{\mathbf{\Delta}}|^2 \right\}. \qquad (5.3)$$

In absence of external sources ($g_{ij} = \delta_{ij}$, $A_\mu = 0$), the saddle point for the pairing field $\overline{\mathbf{\Delta}}$ is given by

$$\overline{\mathbf{\Delta}} = \Delta e^{i\theta} \mathbf{u}, \qquad\qquad \mathbf{u} = (1, \pm i), \qquad (5.4)$$

with $\Delta \geq 0$. $\Delta$ is the magnitude of the order parameter for the symmetry breaking pattern discussed in Section 4, for the specific case of $p$-wave ($l = 1$) superfluidity. While in the normal phase $\Delta = 0$, in the superfluid regime $\Delta \neq 0$. We also define the global phase $\theta$ of the order parameter. The sign $\pm$ in $\mathbf{u}$ distinguishes the two ground states with opposite chiralities, characterized by the angular momentum per particle $s = \pm 1/2$. Both the $U(1)_N$ gauge symmetry and the $SO(2)_R$ rotation symmetry are spontaneously broken, while $\mathbf{p} \cdot \overline{\mathbf{\Delta}}$ remains invariant under the so-called diagonal $U(1)_D$ symmetry. Furthermore, the ground states break down the time reversal ($T$) and parity ($P$) symmetries, while remaining invariant under the $PT$ combination.

Fluctuations of the pairing field around the saddle-point value correspond to the four collective modes of the superfluid. In particular, the gapless phase mode is crucial to preserve the $U(1)_N$ gauge invariance of the theory. The other three modes are gapped and to investigate the theory at the BCS level, we discard their fluctuations while retaining the phase mode to Gaussian (quadratic) order.

For non-vanishing sources, the saddle point value of $\overline{\mathbf{\Delta}}$ depends on both $A_\mu$ and $g_{ij}$. We will now transform the action (5.3) in a manner where the $U(1)_N$ gauge invariance and diffeomorphism invariance are manifest at the mean-field level. First, since under $U(1)_N$ gauge transforms $\psi \to \exp(i\alpha)\psi$ the pairing field has to transform as $\overline{\mathbf{\Delta}} \to \exp(2i\alpha)\overline{\mathbf{\Delta}}$, we now decompose

$$\overline{\mathbf{\Delta}} = \mathbf{\Delta} e^{2i\theta}, \qquad (5.5)$$

with $\theta$ the fluctuating global phase of the pairing field, and perform the unitary transform $\psi \to \exp(i\theta)\psi$. That is equivalent to making the substitution in the action

$$\overline{\mathbf{\Delta}} \to \mathbf{\Delta}, \qquad\qquad A_\mu \to \mathcal{A}_\mu = A_\mu - \partial_\mu \theta, \qquad (5.6)$$

with each term being now manifestly U(1) gauge invariant.

Furthermore, we introduce the a pair of vielbein vectors $e^{ia}$ which satisfy $e^{ia}\delta_{ab}e^{jb} = g^{ij}$. We now decompose $\overline{\Delta}^i = \overline{\Delta}_a e^{ai}$. In curved space, the mean-field configuration is given by $\overline{\Delta}_a \propto u_a = (1, \pm i)$ [1, 34, 56, 57].

After these manipulations, the action reads

$$
S[\psi, \psi^*; \mathcal{A}, g] = \int_x \sqrt{g} \left\{ \psi^* \left[ \partial_\tau - i\mathcal{A}_0 + \frac{g^{ij}(p_i - \mathcal{A}_i)(p_j - \mathcal{A}_j)}{2m} - \mu \right] \psi \right.
$$
$$
\left. - \frac{1}{2} \Delta_a^* e^{ia} (\psi p_i \psi) - \frac{1}{2} \Delta_a e^{ia} (\psi^* p_i \psi^*) \right\} \tag{5.7}
$$

and the BCS approximation is obtained by setting $\Delta$ to its mean-field value, $\Delta_a = \Delta u_a$ with $\Delta > 0$. We have discarded the constant $|\Delta|^2/4\lambda$ contribution to the action which is important to determine the saddle point but doesn't further contribute to the calculation. As the vielbeins transform like vectors under general coordinate transformations, the mean-field action remains diffeomorphism-invariant. Combined with the $U(1)_N$ gauge invariance, this justifies the form of the mean-field action and implies that transport coefficients derived from this theory must satisfy the Ward identities (3.19) and (3.20).

## 5.2 Induced action

The action is quadratic in the fermion fields. We introduce the Nambu spinors

$$
\mathbf{\Psi}_x^\dagger = (\psi_x^*, \psi_x), \qquad\qquad \mathbf{\Psi}_x = (\psi_x, \psi_x^*)^\mathsf{T} \tag{5.8}
$$

to rewrite the action

$$
S[\mathbf{\Psi}, \mathbf{\Psi}^\dagger; \mathcal{A}, h] = -\frac{1}{2} \int_{x, x'} \mathbf{\Psi}_x^\dagger \mathcal{G}_{x, x'}^{-1} \mathbf{\Psi}_{x'}, \tag{5.9}
$$

where $\mathcal{G}^{-1}$ is the inverse Nambu propagator, depending on $\mathcal{A}_\mu$ and $h_{ij} = g_{ij} - \delta_{ij}$. It reads

$$
\mathcal{G}_{x, x'}^{-1} = \sqrt{g} \left\{ -\partial_\tau \sigma^0 + i\mathcal{A}_0 \sigma^z + \frac{1}{2}(\partial_\tau f)\sigma^z - \xi_{\mathbf{p},c}(\mathcal{A}, h) + \Delta e_a^i p_i \tilde{\sigma}^a \right\} \delta(x - x'). \tag{5.10}
$$

In Eq. (5.10), we introduce the shorthand notations $f = \log \sqrt{g}$,

$$
\xi_{\mathbf{p},c}(A, h) = \frac{g^{ij}}{2m} [p_i p_j \sigma^z - (p_i A_j + A_i p_j)\sigma^0 + A_i A_j \sigma^z] - \mu \sigma^0, \tag{5.11}
$$

$\xi_{\mathbf{p},c}(A = 0, h = 0) = \xi_{\mathbf{p}} \sigma^z$, $\xi_{\mathbf{p}} = \mathbf{p}^2/2m - \mu$ and the "twisted" Pauli matrices $\tilde{\sigma}$ defined by $\tilde{\sigma}_y = \mp\sigma_y$, $\tilde{\sigma}_a = \sigma_a$ for $a \neq y$, where $\mp$ is fixed by the chirality of the $p \pm ip$ state.

The inverse bare propagator $\mathcal{G}_0^{-1}$, obtained by dropping $h$ and $\mathcal{A}$, is diagonal in Fourier space,

$$
\mathcal{G}_{0,q}^{-1} = i\omega_n \sigma_0 - \xi_{\mathbf{q}} \sigma_z + \Delta q_a \tilde{\sigma}^a = \begin{pmatrix} i\omega_n - \xi_{\mathbf{q}} & \Delta \mathbf{u} \cdot \mathbf{q} \\ \Delta \mathbf{u}^* \cdot \mathbf{q} & i\omega_n + \xi_{\mathbf{q}} \end{pmatrix}, \tag{5.12}
$$

with $q = (i\omega_n, \mathbf{q})$. Inverting $\mathcal{G}_0^{-1}$ yields

$$
\mathcal{G}_{0,q} = \begin{pmatrix} G_q & F_q \\ F_q^* & -G_{-q} \end{pmatrix}, \tag{5.13}
$$

where we introduce the normal and anomalous Green functions, $G_q$ and $F_q$ respectively,

$$
G_q = -\langle \psi_q \psi_q^* \rangle = -\frac{i\omega_n + \xi_{\mathbf{q}}}{\omega_n^2 + \Delta^2 \mathbf{q}^2 + \xi_{\mathbf{q}}^2}, \qquad F_q = -\langle \psi_q \psi_{-q} \rangle = \frac{\Delta \mathbf{q} \cdot \mathbf{u}}{\omega_n^2 + \Delta^2 \mathbf{q}^2 + \xi_{\mathbf{q}}^2}. \tag{5.14}
$$

For future convenience, we also introduce

$$f_q = \frac{\Delta}{\omega_n^2 + \Delta^2 \mathbf{q}^2 + \xi_{\mathbf{q}}^2} \tag{5.15}$$

such that $F_q = (\mathbf{q} \cdot \mathbf{u}) f_q$.

The next step is to integrate the fermions to obtain an induced action $S[\mathcal{A}, h]$ for the phase mode $\theta$ and the sources $A$ and $h$. Using the standard perturbation theory [13], we expand the inverse propagator in powers of the sources, writing $\mathcal{G}^{-1} = \mathcal{G}_0^{-1} - \Gamma$, with

$$\Gamma_{x,x'} = (1 - \sqrt{g_x}) \mathcal{G}_{0,x,x'}^{-1} - \left\{ \Delta(e_a^i - \delta_a^i) p_i \tilde{\sigma}^a + \left[ i\mathcal{A}_0 + \frac{1}{2} \partial_\tau f \right] \sigma^z - (\xi_{\mathbf{p},c}(A,h) - \xi_{\mathbf{p}} \sigma^z) \right\} \delta(x - x') \tag{5.16}$$

being a vertex representing the coupling of the fermions to the phase mode and the sources.

The effective action then reads up to second order in $h$ and $\mathcal{A}$

$$S[\mathcal{A}, h] = -\operatorname{Tr} \ln(-\mathcal{G}_0^{-1} + \Gamma) = -\operatorname{Tr} \ln(-\mathcal{G}_0^{-1}) + \operatorname{Tr} \mathcal{G}_0 \Gamma + \frac{1}{2} \operatorname{Tr} \mathcal{G}_0 \Gamma \mathcal{G}_0 \Gamma + o(\mathcal{A}^2, h^2), \tag{5.17}$$

where the trace Tr runs over both the spinor indices and space-time coordinates. Due to the presence of both sources $h$ and $A$, keeping track of all terms in the expansion is cumbersome. Because of this we first reorganize Eq. (5.17) in powers of $\mathcal{A}$,

$$S[\mathcal{A}, h] = S_0[h] + \int_x \mathcal{A}_{\mu,x} N_x^\mu[h] + \frac{1}{2} \int_{x,x'} \mathcal{A}_{\mu,x} Q_{x,x'}^{\mu\nu}[h] \mathcal{A}_{\nu,x'}, \tag{5.18}$$

where $S_0[h]$, $N_x^\mu[h]$ and $Q_{x,x'}^{\mu\nu}[h]$ are functionals of the metric $h$ which can be inferred by identifying Eqs. (5.17) and (5.18). By construction $Q_{x,x'}^{\mu\nu}[h]$ is symmetric, $Q_{x,x'}^{\mu\nu}[h] = Q_{x'x}^{\nu\mu}[h]$.

Now we integrate out the phase field $\theta$ to get the induced action

$$\begin{aligned} \mathcal{W}[A, h] = \mathcal{W}_0[h] &+ \int_x A_{\mu,x} N_x^\mu[h] + \frac{1}{2} \int_{x,x'} A_{\mu,x} Q_{x,x'}^{\mu\nu}[h] A_{\nu,x'} \\ &- \frac{1}{2} \int_{x,x'} \partial_{x_\mu} \left( N_x^\mu[h] + \int_y Q_{x,y}^{\mu\nu}[h] A_{\nu,y} \right) [\partial_\mu \partial_\nu Q^{\mu\nu}]_{x,x'}^{-1} \\ &\times \partial_{x'_\mu} \left( N_{x'}^\mu[h] + \int_y Q_{x',y}^{\mu\nu}[h] A_{\nu,y} \right), \end{aligned} \tag{5.19}$$

where $[\partial_\mu \partial_\nu Q^{\mu\nu}]_{x,x'}^{-1}$ is understood as the inverse of $\partial_{y_\mu} \partial_{y'_\nu} Q_{y,y'}^{\mu\nu}$ in the operator sense.

## 5.3 Hall transport coefficients of a chiral superfluid

From the induced action (5.19), the transport coefficients $\sigma^{ij}$, $\eta^{ijkl}$ are deduced by computing functional derivatives of $\mathcal{W}$ wrt $A$ and $h$, see Eqs. (3.7) and (3.8).

### 5.3.1 Conductivity

To determine the conductivity $\sigma^{ij}$, it is sufficient to investigate the theory in flat space $h = 0$. In that case the calculation reduces to that in Ref. [13]; we present its outline in Appendix D.1. To present the results in a compact way, and anticipating the analytic continuation, we make the replacement $iA_\tau \to A_\tau$ and introduce $q_0 = -i\omega_n$ such that $i\mathcal{A}_{0,q} = i(A_{0,q} + i\omega_n \theta_q)$ becomes $\mathcal{A}_{0,q} = A_{0,q} - iq_0 \theta_q$. After that replacement, the induced action for $\mathcal{A}$ becomes

$$S[\mathcal{A}] = \frac{1}{2} \int_q \mathcal{A}_{-q,\mu} Q^{\mu\nu}(q) \mathcal{A}_{q,\nu}, \tag{5.20}$$

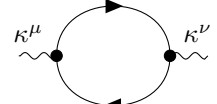

(a) Bubble contribution to $Q^{\mu\nu}(q)$.

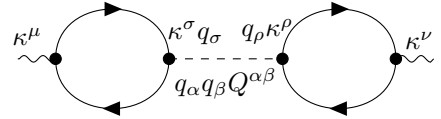

(b) Phase mode contribution to $K^{\mu\nu}(q)$.

Figure 2: Diagrammatic representations of the loop integral contributions to the correlators (a) $Q^{\mu\nu}(q)$ and (b) $K^{\mu\nu}(q)$. The solid line stands for the bare fermion Nambu propagator $\mathcal{G}_0$, the dashed line for the phase mode $\theta$ propagator, the wavy line for the external U(1)$_N$ gauge field $A_\mu$ insertions, and the dots for the interaction vertices $\kappa^\mu$, see Eq. (5.21).

with the correlation (polarization) functions $Q^{\mu\nu}(q)$ defined by

$$Q^{\mu\nu}(q) = \frac{n_0}{m}\delta^{\mu=\nu\neq0} + \frac{1}{2}\int_p \text{tr}[\kappa^\mu_{p,q}\mathcal{G}_{0,p}\kappa^\nu_{p,q}\mathcal{G}_{0,p+q}], \quad \kappa^0_{p,q} = \sigma^z, \quad \kappa^j_{p,q} = \frac{q^j + 2p^j}{2m}\sigma^0, \quad (5.21)$$

with tr denoting the trace only over the internal spinor indices and $\int_p = (2\pi)^{-3}\int \text{d}^2\mathbf{p}\,\text{d}\omega_n$ the summation over both momenta $\mathbf{p}$ and Matsubara frequencies $\text{i}\omega_n$. While the first term on the rhs of Eq. (5.21) is the diamagnetic contribution to the current-current correlation function, the second term is a polarization bubble represented diagramatically in Fig. 2a.

Upon integrating out the phase mode $\theta$ we obtain the induced action for the U(1)$_N$ gauge field $A_\mu$,

$$\mathcal{W}[A] = \frac{1}{2}\int_q A_{-q,\mu}K^{\mu\nu}(q)A_{q,\nu}, \quad (5.22)$$

$$K^{\mu\nu}(q) = Q^{\mu\nu}(q) - \frac{Q^{\mu\rho}(q)q_\rho q_\sigma Q^{\sigma\nu}(q)}{q_\alpha q_\beta Q^{\alpha\beta}(q)}, \quad (5.23)$$

where the phase mode contribution to $K^{\mu\nu}(q)$ is represented in Fig. 2b. Since an arbitrary gauge transform reads $A_\mu \to A_\mu - \text{i}q_\mu\beta(q)$ for some scalar function $\beta$, $q_\mu K^{\mu\nu}(q) = K^{\mu\nu}(q)q_\nu = 0$ enforces the U(1)$_N$ gauge invariance of the induced action. Eq. (5.22) gives

$$\mathcal{W}^{(2,0)ij}(q) = K^{ij}(q) \quad (5.24)$$

from which the conductivity $\sigma^{ij}(\omega,\mathbf{q})$, defined by Eq. (3.7), is deduced after analytic continuation,

$$\sigma^{ij}(\omega,\mathbf{q}) = \frac{1}{\text{i}\omega^+}K^{\text{R},ij}(\omega,\mathbf{q}), \quad (5.25)$$

where $K^{\text{R},ij}(\omega,\mathbf{q}) = K^{ij}(\text{i}\omega_n \to \omega^+,\mathbf{q})$ denotes the retarded part of the correlation function $K^{ij}(\text{i}\omega_n,\mathbf{q})$. The evaluation of the one-loops diagrams contributing to it is done in Appendix D.2. The resulting Hall conductivity, in the small momentum and frequency regime, is

$$\sigma_{\text{H}}(\omega,\mathbf{q}^2) = \frac{sn_0}{2m^2}\frac{-\mathbf{q}^2}{\omega^2 - c_s^2\mathbf{q}^2}, \quad (5.26)$$

where $s = \pm 1/2$ is the angular momentum per particle in the p-wave chiral ground state and $c_s = \sqrt{2\pi n_0/m^2}$ is the speed of sound. This result is in agreement with the effective field theory result (4.3) and the previous microscopic calculation [13].

In the above calculation, two ingredients are necessary to obtain a non-vanishing Hall conductivity. First, preserving the U(1)$_N$ gauge invariance of the theory is crucial. Indeed,

since $Q^{ij}(q) = Q^{ji}(q)$, $\sigma_H$ would vanish if we hadn't kept the phase mode $\theta$. Furthermore, we stress the role of the current-density correlation function, $Q^{0i}(q)$, which is a sum of an even and odd parts, see Eq. (D.11),

$$Q^{0i}(q) = Q^{0i}_e(q) + Q^{0i}_o(q), \qquad Q^{0i}_e(q) = Q^{0i}_e(-q), \qquad Q^{0i}_o(q) = -Q^{0i}_o(-q). \qquad (5.27)$$

In particular, while the density-density $Q^{00}$, current-current $Q^{ij}$ and even current-density $Q^{0i}_e$ correlators are defined in a similar manner as in a non-chiral superfluid (up to the precise form of the gap function), the presence of the odd current-density correlator $Q^{0i}_o$ is only possible due to the time-reversal symmetry breaking. For $\sigma_H(q)$ to be finite, it is necessary to have a nonvanishing $Q^{0i}_o(q)$. Changing the chirality of the ground state from $\mathbf{q} \cdot \mathbf{u} = q_x \pm iq_y$ into $q_x \mp iq_y$ flips the sign of $Q^{0i}_o(q)$ implying $\sigma_H \to -\sigma_H$.

### 5.3.2 Viscosity

To determine the Hall viscosity, it is sufficient to work with the induced action (5.19) with $h_{ij} \neq 0$ and $A_\mu = 0$. Only the fist and last term of Eq. (5.19) remain. Furthermore, as a consequence of Eq. (D.4) $\partial_{x^\mu} N^\mu_x[h] = \mathcal{O}(h)$, so it is enough to expand $N^\mu_x[h]$ and $Q^{\mu\nu}_{x,x'}[h]$ to respectively first and zeroth order in $h$ to get $\mathcal{W}[h]$ to second order in $h$. In particular, $Q^{\mu\nu}_{x,x'}[h = 0]$, i.e. $Q^{\mu\nu}$ evaluated in flat space, has been computed above and is given by Eq. (5.21). The action (5.19) thus simplifies to

$$\mathcal{W}[A,h] = \mathcal{W}_0[h] + \mathcal{W}_\theta[h], \qquad (5.28)$$

where we introduced

$$\mathcal{W}_\theta[h] = -\frac{1}{2}\int_q N^\mu_{-q}[h]\frac{q_\mu q_\nu}{q_\alpha q_\beta Q^{\alpha\beta}(q)}N^\nu_q[h]. \qquad (5.29)$$

The two terms on the rhs of Eq. (5.28) have different origins. $\mathcal{W}_0[h]$ is the contribution to the induced action one would get at the mean-field level; i.e., by discarding the phase mode $\theta$, and we dub it the *pure geometric* contribution. On the other hand, we call $\mathcal{W}_\theta[h]$ the *phase* contribution, as it corresponds to what is obtained by integrating out the phase mode, with two vertices $q_\mu N^\mu_q[h]$ representing an effective interaction between the metric $h$ and the phase mode $\theta$ linked by the inverse Goldstone propagator $q_\alpha q_\beta Q^{\alpha\beta}(q)$.

Since we are interested only in determining $\eta^{(1)}_o$ and $\eta^{(2)}_o$ at leading order in momentum and frequency, we organize the computation accordingly. Dimensional analysis suggests that, at leading order, $\eta^{(1)}_o = \mathcal{O}(|\mathbf{q}|^0)$ and $\eta^{(2)}_o = \mathcal{O}((c_s^{-2}\omega^2 - \mathbf{q}^2)^{-1})$, an intuition confirmed by the effective field theory calculation [see Eq. (4.4), and [34] for the calculation up to the next-to-leading order]. Either term $\mathcal{W}_0[h]$ and $\mathcal{W}_\theta[h]$ brings a different contribution to the viscosity tensor. The one-loop integrals appearing in $\mathcal{W}_0[h]$ and the vertices $q_\mu N^\mu_q[h]$ are all regular in the infrared limit $\mathbf{q} \to 0$, $\omega \to 0$ and thus, the only way a term of order $\mathcal{O}(|\mathbf{q}|^{-2})$ can appear in the calculation is through the inverse Goldstone propagator $\sim (c_s^{-2}\omega^2 - \mathbf{q}^2)$ from the phase contribution. Hence, the leading contribution to $\eta^{(2)}_o$ is entirely fixed by $\mathcal{W}_\theta[h]$. Conversely, as the Goldstone propagator does not appear in $\eta^{(1)}_o$ at the leading order, $\eta^{(1)}_o$ is determined by $\mathcal{W}_0[h]$. We now compute each contribution to the viscosity separately.

**Contribution from the pure geometric part**    Here we start from the fermionic action defined by the inverse propagator (5.10), discarding for now the phase mode $\theta$. We work at vanishing $\mathrm{U}(1)_N$ gauge field, i.e. $\mathcal{A} = 0$. The fermions are integrated out following Eq. (5.17), where $\Gamma$ is

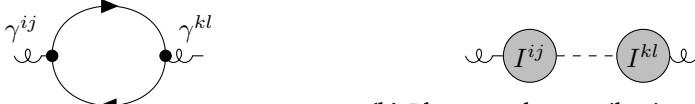

(a) Bubble contribution to $\mathcal{W}_0[h]$.

(b) Phase mode contribution to $\mathcal{W}_\theta[h]$.

Figure 3: Diagrammatic representation of the loop integrals contributions to the pure geometric (a) and phase parts (b) of the induced action. The curly line stands for the metric $h_{ij}$ insertions, the dots for the fermion-metric interaction vertices $\gamma^{ij}$ [Eq. (5.32)] and the grey blobs for the phase-metric interaction vertices $I^{ij}$ [Eq. (5.36)].

given by Eq. (5.16) with $\mathcal{A}$ set to zero. The detailed calculation is presented in Appendix D.3. The end result is

$$\mathcal{W}_0[h] = \frac{1}{2}\int_q h_{-q,ij}R^{ij,kl}(q)h_{q,kl},\tag{5.30}$$

with

$$R^{ij,kl}(q) = \frac{1}{2}\int_p \mathrm{tr}[\mathcal{G}_{0,p}\gamma^{ij}_{p,-p-q}\mathcal{G}_{0,p+q}\gamma^{kl}_{p+q,-p}],\tag{5.31}$$

$$\gamma^{ij}_{q,-q'} = \frac{1}{2}\delta^{ij}\left((2q_0 - q'_0)\sigma^0 - \frac{\mathbf{q}\cdot\mathbf{q}'}{2m}\sigma^z + \Delta q'_k\tilde{\sigma}^k\right) - \frac{q^{(i}q'^{j)}}{2m}\sigma^z + \frac{1}{2}\Delta q'^{(i}\tilde{\sigma}^{j)}.\tag{5.32}$$

The tensor $R^{ij,kl}(q)$ is defined by a one-loop integral analogous to the polarization bubble $Q^{ij}$ [Eq. (5.21)] that is relevant in the calculation of the conductivity. It is represented diagramatically in Fig. 3a. From Eq. (3.7), it is related to the odd viscosity through

$$\eta_o^{ijkl}(\omega,\mathbf{q}) = \frac{4}{i\omega^+}R^{\mathrm{R},ijkl}(\omega,\mathbf{q}),\tag{5.33}$$

where $R^{\mathrm{R},ijkl}(\omega,\mathbf{q}) = R^{\mathrm{R},ijkl}(i\omega \to \omega^+,\mathbf{q})$. The odd viscosities are obtained by projecting onto the odd tensors $\sigma^{xz}$ for $\eta_o^{(1)}$ and $\sigma^{0x}$ or $\sigma^{0z}$ for $\eta_o^{(2)}$, as done in Appendix D.4. At small frequency and momentum, one gets

$$\eta_o^{(1)}(\omega,\mathbf{q}^2) = \frac{sn_0}{2} + \mathcal{O}(\omega^2,\mathbf{q}^2),\tag{5.34}$$

while the contribution to $\eta_o^{(2)}$ is of order $\mathcal{O}(|\mathbf{q}|^0)$, i.e. is subleading.

**Contribution from the phase**  We now compute the contributions to the viscosity tensor that originate from the phase mode. The linear term $N_q^\mu[h]$ is determined to leading order in $h_{ij}$ in Appendix D.5, yielding

$$q_\mu N_q^\mu[h] = h_{ij,q}I_q^{ij},\tag{5.35}$$

$$I_q^{ij} = -\frac{1}{2}\mathrm{tr}\int_p \mathcal{G}_{0,p}\left[q_0\sigma^z + \frac{(2\mathbf{p}+\mathbf{q})\cdot\mathbf{q}}{2m}\sigma^0\right]\mathcal{G}_{0,p+q}\left[\frac{(p+q)^{(i}p^{j)}}{2m}\sigma^z + \frac{1}{2}\Delta\tilde{\sigma}^{(i}p^{j)}\right].\tag{5.36}$$

The resulting contribution to the viscosity tensor $\eta^{ijkl}$ is given by

$$\eta_o^{ijkl}(\omega,\mathbf{q}) = \frac{4}{i\omega^+}S^{\mathrm{R},ijkl}(\omega,\mathbf{q}),\tag{5.37}$$

with $S^{\mathrm{R},ijkl}(\omega,\mathbf{q})$ the retarded part of

$$S^{ijkl}(q) = -\frac{I^{ij}_{-q}I^{kl}_{q}}{q_\alpha q_\beta Q^{\alpha\beta}(q)}, \tag{5.38}$$

represented diagramatically in Fig. 3b.

The projection of the corresponding viscosity tensor on $\sigma^{ab}$ matrices is done in Appendix D.5. For $\eta^{(1)}_{\mathrm{o}}$, the projection vanishes, hence the contribution from the phase mode to $\eta^{(1)}_{\mathrm{o}}$ is at most of order $\mathcal{O}(|\mathbf{q}|^4/(c_s^{-2}\omega^2-\mathbf{q}^2))$. For $\eta^{(2)}_{\mathrm{o}}$, the direct calculation yields

$$\eta^{(2)}_{\mathrm{o}}(\omega,\mathbf{q}^2) = -\frac{sn_0}{2}\frac{c_s^2}{\omega^2-c_s^2\mathbf{q}^2}. \tag{5.39}$$

Having determined the odd conductivity (5.26) as well as the two components of the odd viscosity tensor, (5.34) and (5.39), one checks that the Ward identity (3.20) is satisfied. Contrary to the case of the conductivity, where the incorporation of the phase mode is crucial to preserve the $\mathrm{U}(1)_N$ gauge invariance and get the correct result for the associated transport coefficient $\sigma^{\mathrm{H}}$, it is not obvious *a priori* whether including the phase mode is important or not to obtain the viscosity tensor. This is reflected in the calculation as the phase mode does not affect the value of $\eta^{(1)}_{\mathrm{o}}$ but is crucial to obtain $\eta^{(2)}_{\mathrm{o}}$. We notice that only by going beyond mean-field is the Ward identity (3.20) fulfilled, as expected since the mean-field theory breaks down $\mathrm{U}(1)_N$ gauge invariance which the identity relies on.

## 5.4 Inclusion of Coulomb and non-local interactions

In this section, we now additionally incorporate a non-local Coulomb interaction between the fermions, for which the corresponding Euclidean action reads in flat space

$$S_{\mathrm{C,f}}[\psi,\psi^*] = \frac{1}{2}\int_{x,x'}(\psi^\dagger\psi-\bar{n})V(\mathbf{r}-\mathbf{r}')(\psi^\dagger\psi-\bar{n}), \tag{5.40}$$

with $V(\mathbf{r})$ the interaction potential, $e$ the electric charge and $\bar{n}$ the background density. We consider potentials satisfying $V(\mathbf{q}) \sim |\mathbf{q}|^{-\alpha}$ at long distances ($\mathbf{q} \to 0$), with $0 \le \alpha < 2$. The case $\alpha = 0$ corresponds to of short-ranged interactions, $\alpha = 1$ to a mixed-dimensional Coulomb interaction, and $\alpha = 2$ to an intrictically two-dimensional Coulomb potential, see Section 4.3 for a more thourough discussion.

The density-density interaction in Eq. (5.40) is decoupled by means of a Hubbard-Stratanovitch transform, with an auxiliary field $\chi$, yielding

$$S_{\mathrm{C,f}}[\psi,\psi^*,\chi] = \frac{1}{2}\int_x \chi V^{-1}\chi + \int_x(\psi^*\psi-\bar{n})(-\mathrm{i}\chi), \tag{5.41}$$

where $V^{-1} = (-\boldsymbol{\nabla}^2)^{\alpha/2}/e^2$ is the inverse propagator for the Coulomb field. The expression (5.41) allows to generalize the theory to arbitrary curved space[14]

$$S_{\mathrm{C}}[\psi,\psi^*,\chi;g] = \frac{1}{2}\int_x \sqrt{g}\chi V^{-1}\chi + \int_x \sqrt{g}(\psi^*\psi-\bar{n})(-\mathrm{i}\chi), \tag{5.42}$$

with the Laplace operator in $V^{-1}$ replaced by the covariant Laplacian $g^{ij}\nabla_i\nabla_j$.

---

[14]To formulate the theory in curved space starting from the action (5.40), one would need to replace the distance $\mathbf{r}-\mathbf{r}'$ by the geodesic distance [39]

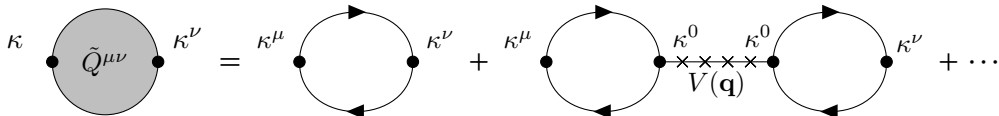

Figure 4: Renormalization of the polarization bubbles $Q^{\mu\nu}(q)$ due to the Coulomb interaction. The crossed line denotes the Coulomb potential $V(\mathbf{q})$ which only couples to fermions through the density vertices $\kappa^0$.

The Coulomb interaction couples to the fermions like the time-component of the $U(1)_N$ gauge field. Hence, the total action $S_T = S + S_C$ reads

$$S_T[\Psi, \Psi^\dagger; \tilde{\mathcal{A}}, h] = S[\Psi, \Psi^\dagger; \tilde{\mathcal{A}}, h] + \frac{1}{2}\int_x \sqrt{g}\,\chi V^{-1}\chi + i\bar{n}\int_x \sqrt{g}\,\chi, \qquad (5.43)$$

where $S$ is the action (5.9) and we introduced $\tilde{\mathcal{A}}_0 = \tilde{\mathcal{A}}_0 + \chi$, $\tilde{\mathcal{A}}_i = \mathcal{A}_i$. We integrate out the fermions following Eqs. (5.17) and (5.18) to obtain the effective action for $\phi$, $\chi$, $A$, and $h$ to quadratic order in fields and sources,

$$S_T[\tilde{\mathcal{A}}, h] = \mathcal{W}_0[h] + \int_x \tilde{\mathcal{A}}_{\mu,x} N_x^\mu[h] + \frac{1}{2}\int_{x,x'} \tilde{\mathcal{A}}_{\mu,x} Q_{x,x'}^{\mu\nu}[h] \tilde{\mathcal{A}}_{\nu,x'}$$

$$+ i\bar{n}\int_x \sqrt{g}\,\chi + \frac{1}{2}\int_x \sqrt{g}\,\chi V^{-1}\chi. \qquad (5.44)$$

At this stage, we integrate out the Coulomb interaction. The linear terms in $\chi$ proportional to $N_x^\mu[h]$ and to $\bar{n}$ in Eq. (5.44) compensate each other to ensure charge neutrality and the effective action for $\mathcal{A}$ and $h$ reads

$$S[\mathcal{A}, h] = \mathcal{W}_0[h] + \int_x \mathcal{A}_{\mu,x} N_x^\mu[h] + \frac{1}{2}\int_{x,x'} \mathcal{A}_{\mu,x} \tilde{Q}_{x,x'}^{\mu\nu}[h] \mathcal{A}_{\nu,x'}, \qquad (5.45)$$

where $\tilde{Q}[h]$ is defined by

$$\tilde{Q}_{\mu\nu}[h] = Q_{\mu\nu}[h] + Q_{\mu 0}[h]\frac{V}{\sqrt{g} - V Q_{00}} Q_{0\nu}[h], \qquad (5.46)$$

with the products defined in the operator sense. In the specific case of flat space, $\tilde{Q}_{\mu\nu} = \tilde{Q}_{\mu\nu}[h=0]$ reduces to

$$\tilde{Q}_{00}(q) = \frac{Q_{00}(q)}{1 - V(\mathbf{q})Q_{00}(q)}, \quad \tilde{Q}_{j0}(q) = \frac{Q_{j0}(q)}{1 - V(\mathbf{q})Q_{00}(q)}, \qquad (5.47)$$

$$\tilde{Q}_{ij}(q) = Q_{ij}(q) + \frac{V(\mathbf{q})Q_{i0}(q)Q_{0j}(q)}{1 - V(\mathbf{q})Q_{00}(q)}. \qquad (5.48)$$

Expanding the Coulomb field to quadratic order and integrating it out is equivalent to the random phase approximation (RPA) and expressions (5.46) to (5.48) can equivalently be obtained through resummation of the bubble diagrams illustrated in Fig. 4.

Comparing the actions (5.18) and (5.45), the only difference is that the kernel $Q_{\mu\nu}[h]$ has been replaced by $\tilde{Q}_{\mu\nu}[h]$. It is straightforward now to extract the Hall transport coefficients. Following Section 5.3.1, one finds the induced action for the $U(1)_N$ gauge field in flat space to be

$$\mathcal{W}[A] = \frac{1}{2}\int_q A_{-q,\mu}\tilde{K}^{\mu\nu}(q)A_{q,\nu}, \qquad \tilde{K}^{\mu\nu}(q) = \tilde{Q}^{\mu\nu}(q) - \frac{\tilde{Q}^{\mu\rho}(q)q_\rho q_\sigma \tilde{Q}^{\sigma\nu}(q)}{q_\alpha q_\beta \tilde{Q}^{\alpha\beta}(q)}, \qquad (5.49)$$

yielding the Hall conductivity

$$\sigma_{\text{H}}(\omega, \mathbf{q}^2) = \frac{sn_0}{2m^2} \frac{-\mathbf{q}^2}{\omega^2 - \omega_p^2 |\mathbf{q}|^{2-\alpha} - c_s^2 \mathbf{q}^2}. \tag{5.50}$$

As for the viscosity, we follow Section 5.3.2. The calculation of $\eta_o^{(1)}$ does not involve $\tilde{Q}$ and $\eta_o^{(1)}$ is thus unaffected by the interaction. On the other hand, $\eta_o^{(2)}$ is related to

$$\tilde{S}^{ijkl}(q) = -\frac{I_{-q}^{ij} I_q^{kl}}{q_\alpha q_\beta \tilde{Q}^{\alpha\beta}(q)}, \tag{5.51}$$

through Eq. (5.37), and thus becomes

$$\eta_o^{(2)}(\omega, \mathbf{q}^2) = -\frac{sn_0}{2} \frac{1}{|\mathbf{q}|^\alpha} \frac{c_s^2 |\mathbf{q}|^\alpha + \omega_p^2}{\omega^2 - \omega_p^2 |\mathbf{q}|^{2-\alpha} - c_s^2 \mathbf{q}^2} \tag{5.52}$$

in presence of the long-range interaction $V(\mathbf{r}) \sim |\mathbf{r}|^{2-\alpha}$.

The Hall conductivity and viscosities obtained here agree with what was found in Sections 4.2 and 4.3 from the effective field theory approach.

# 6 Discussion and outlook

In this paper we computed electromagnetic and geometric linear responses in non-relativistic two-dimensional chiral superconductors, where in addition to short-range attractive interactions that lead to chiral pairing, elementary fermions interact via an instantaneous long-range Coulomb potential. For the two-dimensional logarithmic Coulomb interaction we found that the homogeneous $\mathbf{q} \to 0$ limit of the Hall viscosity tensor $\eta_o$ is ill-defined because the result depends on the direction of the momentum vector $\mathbf{q}$. We believe that this peculiar behavior is an artifact of the instantaneous nature of the Coulomb potential. It is expected that the problematic $\mathbf{q}^2$ denominator of the component $\eta_o^{(2)}$ in Eq. (4.15) is replaced by $\mathbf{q}^2 - \omega^2/c^2$ after the Coulomb potential is replaced by a retarded electromagnetic interaction that propagates with a finite speed of light $c$. In this way at a finite frequency $\omega$ the $1/\mathbf{q}^2$ singularity will be regularized. To clarify this issue in a future work we are planning to compute the Hall viscosity and conductivity in the Lorentz-invariant chirally paired model [58] coupled to the Maxwell electromagnetism. This approach can also shed new light on the geometric Meissner effect [52] and the geometric induction [59] in chiral superconductors.

It would be interesting to extend the ideas presented in this paper to non-abelian quantum Hall states (such as Pfaffian, anti-Pfaffian, particle-hole symmetric Pfaffian) which can be viewed as chiral paired states of composite fermions [1,44] that couple to a dynamical $2+1$ dimensional abelian gauge field.

Inspired by recent work on viscoelastic linear response of anisotropic systems [60,61], it would be interesting and straightforward to extend this work to anisotropic chiral superconductors.

# Acknowledgements

We thank Ady Stern and Carlos Hoyos for productive discussions and comments on the manuscript. We acknowledge fruitful discussions with Nicolas Dupuis, Dam Thanh Son, Anton Souslov

and Wilhelm Zwerger. OG acknowledges support from the Israel Science Foundation (ISF), the Deutsche Forschungsgemeinschaft (DFG, CRC/Transregio 183, EI 519/7-1), and the European Research Council (ERC, Project LEGOTOP). The work of SM is funded by the Deutsche Forschungsgemeinschaft (DFG, German Research Foundation) under Emmy Noether Programme grant no. MO 3013/1-1 and under Germany's Excellence Strategy - EXC-2111 - 390814868.

## A  Ward identities

In this Appendix we derive the continuity equations (3.17) and (3.18) and the Ward identity (3.19). Our starting point is Eq. (3.16) which implies that for the infinitesimal transforms defined by (3.13) to (3.15)

$$\int_x \delta A_\mu(x) \mathcal{W}^{(1,0)\mu}[x;A,g] + \delta g_{ij}(x) \mathcal{W}^{(0,1)ij}[x;A,g] = 0. \tag{A.1}$$

This holds for any choice of $\alpha(x)$, $\xi^i(x)$ and thus

$$\partial_\mu \mathcal{W}^{(1,0)\mu} = 0, \tag{A.2}$$

$$m\partial_t(\mathcal{W}_k^{(1,0)}) - [(\partial_j g_{ik} + \partial_i g_{jk} - \partial_k g_{ij})\mathcal{W}^{(0,1)ij} + (g_{ik}\partial_j \mathcal{W}^{(0,1)ij} + g_{jk}\partial_i \mathcal{W}^{(0,1)ij})]$$
$$= (\partial_\mu A_k - \partial_k A_\mu)\mathcal{W}^{(1,0)\mu}, \tag{A.3}$$

where every quantity is evaluated at arbitrary spacetime point and for any value of the sources and $k$ is an arbitrary space index. The indices are raised and lowered using the spatial metric tensor $g$. Using now Eq. (3.3) one gets the continuity equations, Eqs. (3.17) and (3.18).

By taking a further derivative of Eq. (A.3) wrt $A_\nu$ and setting $A = 0$, $g = \delta$ we get in Fourier space

$$-m\omega^+ \mathcal{W}^{(2,0)k\nu}(q) - [q_j \mathcal{W}^{(1,1)\nu,kj}(-q) + q_i \mathcal{W}^{(1,1)\nu,ik}(-q)]$$
$$+ \frac{1}{\sqrt{V}}\{q^k \mathcal{W}^{(1,0)\nu}(q=0) + \delta_\nu^k[\omega^+ \mathcal{W}^{(1,0)t}(q=0) - q_i \mathcal{W}^{(1,0)i}(q=0)]\} = 0. \tag{A.4}$$

We also differentiate Eq. (A.3) wrt $g_{ab}$. This gives[15]

$$-m\omega^+ \mathcal{W}^{(1,1)k,ab}(q) - 2q_j \mathcal{W}^{(0,2)(kj),ab}(q)$$
$$= \frac{1}{\sqrt{V}}\{\delta^{kb}q_j \mathcal{W}^{(0,1)(aj)}(q=0) + \delta^{ka}q_j \mathcal{W}^{(0,1)(bj)}(q=0) - q^k \mathcal{W}^{(0,1)ab}(q=0)\}. \tag{A.5}$$

We contract Eq. (A.5) with $q_a$ and cancel the resulting $q_a \mathcal{W}^{(1,1)k,ab}$ term with the one appearing in Eq. (A.4) to get

$$m^2(\omega^+)^2 \mathcal{W}^{(2,0)ij}(q) = 4q_k q_l \mathcal{W}^{(0,2)ikjl}(q) + \frac{1}{\sqrt{V}}\{2\delta^{ij}q_k q_l \mathcal{W}^{(0,1)kl}(q=0)$$
$$-m\omega^+[q^i \mathcal{W}^{(1,0)j}(q=0) + \delta^{ij}(\omega^+ \mathcal{W}^{(1,0)t}(q=0) - q_k \mathcal{W}^{(1,0)k}(q=0))]\}. \tag{A.6}$$

The second derivatives of the induced action $\mathcal{W}$ can be related to physical observables via Eqs. (3.7) and (3.8). For vanishing sources

$$\langle J^\mu \rangle = 0, \qquad\qquad \langle T^{ij} \rangle = P\delta^{ij} \tag{A.7}$$

---

[15]To get a fully symmetric expression we remark that $\mathcal{W}^{(0,1)ij} = \mathcal{W}^{(0,1)(ij)} = (\mathcal{W}^{(0,1)ij} + \mathcal{W}^{(0,1)ji})/2$ and use $\delta g^{ab}/\delta g^{ij} = (\delta_i^a \delta_j^b + \delta_j^a \delta_i^b)/2$.

and to leading order in momentum

$$\lambda^{ijkl}(\mathbf{q}, \omega) = P(\delta^{ik}\delta^{jl} + \delta^{il}\delta^{jk}) + \kappa^{-1}\delta^{ij}\delta^{kl} + \mathcal{O}(|\mathbf{q}|), \tag{A.8}$$

with $P$ the pressure and $\kappa^{-1} = -V\left(\partial P/\partial V\right)_{S,N}$ the inverse compressibility. Using these results in Eq. (A.6) yields the Ward identity (3.19).

## B  Gaussian functional integration in curved space

Here we provide some details on how Gaussian functional integration is performed in the presence of a general background metric $g_{ij}(t, \mathbf{x})$.

Consider a $d$-dimensional Gaussian scalar theory defined in a general coordinate system by the Euclidean action

$$S_{\mathrm{E}} = \int_x \sqrt{g}\left[\frac{1}{2}\varphi a\varphi + b\varphi\right], \tag{B.1}$$

where $g = \det g_{ij}$, $a$ is a symmetric operator and $b$ is some given scalar function of space and time.

The general-coordinate invariant inner product in the space of scalar fields is given by

$$\langle\varphi_1|\varphi_2\rangle = \int_x \sqrt{g}\varphi_1\varphi_2, \tag{B.2}$$

where the volume element $\sqrt{g(t, \mathbf{x})}$ can be viewed as a (diagonal) metric tensor in the space of scalar fields. It follows that the general-coordinate invariant functional measure is [62–64]

$$\mathcal{D}[\varphi] = \prod_{\mathbf{x}} g^{1/4}(\mathbf{x})\mathrm{d}\varphi(\mathbf{x}). \tag{B.3}$$

As a result, the Gaussian functional integral in Euclidean time can be performed as follows,

$$
\begin{aligned}
\int \mathcal{D}[\varphi]\mathrm{e}^{-\int_x \sqrt{g}\left[\frac{1}{2}\varphi a\varphi + b\varphi\right]} &= \frac{\prod_{\mathbf{x}} g^{1/4}}{\sqrt{\det\left(\sqrt{g}a/2\pi\right)}}\mathrm{e}^{\frac{1}{2}\int_x\left[\sqrt{g}b\left(\sqrt{g}a\right)^{-1}\sqrt{g}b\right]} \\
&= \frac{1}{\sqrt{\det(a/2\pi)}}\mathrm{e}^{\frac{1}{2}\int_x \sqrt{g}ba^{-1}b}.
\end{aligned}
\tag{B.4}
$$

## C  Linear response from EFT of two-dimensional chiral superconductors

In this Appendix we compute to leading order in derivatives and within RPA the induced action resulting from the EFT (4.16) which describes a chiral superconductor, where charged fermions interact via a two-body potential whose form in Fourier space is $V(\mathbf{q}) \sim |\mathbf{q}|^{-\alpha}$. We consider the range $0 \leq \alpha \leq 2$, which includes short range interactions at $\alpha = 0$, the $3d$ Coulomb interaction $V(\mathbf{r}) \sim 1/|\mathbf{r}|$ at $\alpha = 1$, and the $2d$ Coulomb interaction $V(\mathbf{r}) \sim \log|\mathbf{r}|$ at $\alpha = 2$.

In Appendix C.1 we do so using a formalism based on the Legendre transform of the pressure functional. That calculation, complementary to what is done in Sections 4.2 and 4.3, unifies the superconductor EFTs (4.5) and (4.16) with the superfluid EFT (4.1) by introducing the renormalized speed of sound. We then derive and analyze the resulting linear response functions in Appendix C.2.

## C.1 Legendre transform interpretation of the effective theory and renormalized speed of sound

### C.1.1 Generalities

We start by generalizing Eq. (4.16) to

$$S = \int_x \sqrt{g} \left[ P(X - \chi) + n_0 \chi + \frac{1}{2} \chi \frac{1}{V(-\boldsymbol{\nabla}^2)} \chi \right], \tag{C.1}$$

where $V(\mathbf{q}^2)$ is the Fourier transform of a density-density interaction between microscopic fermions. We then rewrite $P(X) = X\rho - \varepsilon(\rho)$, where $\rho$ is a new dynamical field which physically corresponds to the density, and $\varepsilon = \mathcal{L}\{P\}$ is the Legendre transform of $P$, the internal energy density of the superfluid [14]. Eq. (C.1) can then be written as

$$S = \int_x \sqrt{g} \left[ X\rho - \varepsilon(\rho) - (\rho - n_0)\chi + \frac{1}{2} \chi \frac{1}{V(-\boldsymbol{\nabla}^2)} \chi \right]. \tag{C.2}$$

Since $\chi$ appears quadratically it can be integrated out exactly, leading to

$$S = \int_x \sqrt{g} (\rho X - \tilde{\varepsilon}[\rho]), \tag{C.3}$$

where $\tilde{\varepsilon}[\rho] = \varepsilon(\rho) - (\rho - n_0) V(-\boldsymbol{\nabla}^2)(\rho - n_0)/2$ is the internal energy density of the superfluid, supplemented by the interaction energy. The square brackets indicate that $\tilde{\varepsilon}$ is generally a functional, as opposed to the function $\varepsilon$. Integrating out $\rho$ is now equivalent to a (functional) Legendre transform from $\tilde{\varepsilon}$ to $\tilde{P} = \mathcal{L}\{\tilde{\varepsilon}\}$,[16]

$$S = \int_x \sqrt{g} \tilde{P}[X]. \tag{C.4}$$

The superconductor EFT (4.16) is therefore equivalent to the superfluid EFT (4.1), with a renormalized pressure functional $\tilde{P}[X]$ which simply accounts for the additional interaction energy $V(\mathbf{q}^2)$ of microscopic fermions.

### C.1.2 Quadratic pressure functional and speed of sound

Let us now assume that the pressure functional $P(X)$ is quadratic,

$$P(X) = P_0 + n_0(X - \mu) + \frac{1}{2} \frac{n_0}{mc_s^2}(X - \mu)^2. \tag{C.5}$$

In this case the Legendre transform $\tilde{P}[X]$ can be computed explicitly and one finds

$$\tilde{P}[X] = P_0 + n_0(X - \mu) + \frac{1}{2} \frac{n_0}{mc_s^2}(X - \mu) \frac{1}{1 + n_0 V(-\boldsymbol{\nabla}^2)/mc_s^2}(X - \mu). \tag{C.6}$$

Comparing with Eq. (C.5), we see that the only effect of the interaction $V(\mathbf{q}^2)$ is to renormalize the speed of sound, $c_s^2 \to \tilde{c}_s^2 = c_s^2 + n_0 V(-\boldsymbol{\nabla}^2)/m$, where $\tilde{c}_s$ has been promoted from a number to an operator. This reads in Fourier space

$$c_s^2 \to \tilde{c}_s^2(\mathbf{q}^2) = c_s^2 + \omega_p^2 |\mathbf{q}|^{-\alpha} \tag{C.7}$$

---

[16]Note that the Legendre transform is an involution, $\mathcal{L}^{-1} = \mathcal{L}$.

for $V(\mathbf{q}^2) = e^2|\mathbf{q}|^{-\alpha}$. Thus, for quadratic $P$, the superconductor EFT is identical to the superfluid EFT with a renormalized speed of sound. Note that this renormalization is trivial for the contact interaction $\alpha = 0$, since in this case $c_s$ is independent of $\mathbf{q}$, as in the superfluid. Using the thermodynamic expression $K^{-1} = n^{-1}(\delta n/\delta \tilde{P})_T = n_0 m c_s^2$, Eq. (C.7) implies the (zero temperature) inverse compressibility

$$K^{-1}(\mathbf{q}) = n_0 m(c_s^2 + \omega_p^2|\mathbf{q}|^{-\alpha}). \tag{C.8}$$

In particular, for all $\alpha > 0$ the superconductor is incompressible with respect to a uniform compression, $K(q = 0) = 0$. In Appendix C.2 we will use Eq. (C.7) to deduce the induced action of the superconductor from that of the superfluid.

### C.1.3 Approximations: linear response, derivative expansion, and RPA

The pressure functional $P$ is not quadratic in general, and in particular in the microscopic model of Section 5. However, for the purpose of computing linear response functions, to lowest order in derivatives, and within RPA, only an expansion of $P$ to quadratic order (C.5) around $\mu$ is sufficient, because the effective theory Eq. (C.1) can be expanded to second order in all fields. Indeed, linear response is extracted from the quadratic expansion in background fields. To lowest order in derivatives, diagrams with the Goldstone field $\varphi$ running in loops can be neglected [34,53], which amounts to a quadratic expansion in $\varphi$. Finally, the RPA amounts to neglecting diagrams with $\chi$ running in loops, i.e a quadratic expansion in $\chi$. Explicitly, to second order in all fields Eq. (C.1) reduces to

$$
\begin{aligned}
S = \int_x \sqrt{g} \Big\{ &P_0 - n_0 D_t \varphi + \frac{n_0}{2m}[c_s^{-2}(D_t \varphi)^2 - (\mathbf{D}\varphi)^2] \\
&+ \frac{n_0}{m c_s^2}\Big[\chi D_t \varphi + \frac{1}{2}\chi(c_s^2 \omega_p^{-2}(-\boldsymbol{\nabla}^2)^\alpha + 1)\chi\Big] \Big\},
\end{aligned} \tag{C.9}
$$

and integrating over $\chi$ leads to the speed of sound renormalization

$$S = \int_x \sqrt{g} \Big\{ P_0 - n_0 D_t \varphi + \frac{n_0}{2m}[\tilde{c}_s^{-2}(D_t \varphi)^2 - (\mathbf{D}\varphi)^2] \Big\}, \tag{C.10}$$

which is nothing but the action (4.9). We note that space and time derivatives are counted equally, and that the leading derivative corrections to (C.10) are due to second order terms that add to $P(X)$ in the superfluid EFT [34,53], rather than $\varphi$ loops. As a result, the linear response functions obtained in Appendix C.2 below, which can formally be expanded to infinite order in derivatives, receive corrections at order $L + 2$, where $L$ is their leading order in derivatives. In particular, $\sigma_{\mathrm{H}}$ in Eq. (C.13) below, is of order $\alpha$, and receives derivative corrections at order $\alpha + 2$. Additionally, $\eta_{\mathrm{o}}^{(2)}$ in Eq. (C.19) is of leading order $-2$, and receives corrections at zeroth order.

## C.2 Induced action and linear response

Since the quadratic effective action (C.10) is identical to that of the superfluid under the replacement $c_s \to \tilde{c}_s$, the quadratic induced action in the superconductor can be directly obtained from that of the superfluid by applying the same replacement. The induced action in chiral superfluids was computed and analyzed in Refs. [14, 34], and below we use the results of Appendix D of Ref. [34]. Replacing $c_s \to \tilde{c}_s$, the induced action of the chiral superconductor

expanded around flat space is we obtain

$$
\mathcal{W}[g,A] = \int_x \Big[ 2P_0 h - n_0 A_t
$$
$$
+ \frac{1}{2} \frac{n_0}{m} \frac{\tilde{c}_s^2 B^2 - E^2 + (is/m)E^i q_i B - (s^2/4m^2)\mathbf{q}^2 B^2}{\omega^2 - \tilde{c}_s \mathbf{q}^2}
$$
$$
+ 2n_0 \frac{\tilde{c}_s^2 h[iq_i E^i + (s/2m)\mathbf{q}^2 B] - m\tilde{c}_s^2 \omega^2 h^2}{\omega^2 - \tilde{c}_s^2 \mathbf{q}^2} \Big],
\tag{C.11}
$$

where $h_{ij} = g_{ij} - \delta_{ij}$ and $h = h_i^i$. This expression encodes the entire linear response of the chiral superconductor in flat space within RPA and to the lowest order in derivatives, as we now discuss.

First, note that the terms in $\mathcal{W}$, and the corresponding linear response functions, can be split into three groups, according to their dependence on $\tilde{c}_s$. The terms in the first group are independent of $\tilde{c}_s$, and accordingly, they are unaffected by the additional fermion interaction $V$ that distinguishes the superconductor from the superfluid. These appear in the first line of (C.11), and include the ground state pressure $P_0$ and density $n_0$, as well the first odd viscosity $\eta_o^{(1)} = sn_0/2$ [Eq. (4.19)].

The second group of terms in $\mathcal{W}$ includes those where $\tilde{c}_s$ appears only in the denominator, and include the terms $-E^2 + (is/m)E^i q_i B$ be derived from $-E^2 + (is/m)E^i q_i B$ in the second line of (C.11), which encode the longitudinal and Hall conductivities, $\sigma$ and $\sigma_H$. To write down the conductivities we will use the explicit plasmon propagator

$$
\frac{1}{\omega^2 - \tilde{c}_s^2(\mathbf{q}^2)\mathbf{q}^2} = \frac{1}{\omega^2 - c_s^2 \mathbf{q}^2 - \omega_p^2 |\mathbf{q}|^{2-\alpha}},
\tag{C.12}
$$

which is gapped in the case $\alpha = 2$, but has a gapless dispersion relation $\omega \sim |\mathbf{q}|^{1-\alpha/2}$ for $0 \leq \alpha < 2$. We then find the longitudinal and Hall [Eq. (4.18)] conductivities,

$$
\sigma(\omega,\mathbf{q}) = \frac{n_0}{m} \frac{i\omega}{\omega^2 - |\mathbf{q}|^{2-\alpha}\omega_p^2 - c_s^2\mathbf{q}^2}, \qquad \sigma_H(\omega,\mathbf{q}) = \frac{sn_0}{2m^2} \frac{-\mathbf{q}^2}{\omega^2 - |\mathbf{q}|^{2-\alpha}\omega_p^2 - c_s^2\mathbf{q}^2}.
\tag{C.13}
$$

Note that here and below, the superfluid expressions can be obtained by setting $\omega_p = 0$. We can also extract the density-density response $\chi_{nn} = iq^2\sigma/\omega$ and verify using Eq. (C.8), the compressibility sum-rule $K^{-1}(\mathbf{q}) = n_0^2/\chi_{nn}(\omega = 0, \mathbf{q})$, which follows from the thermodynamic identity $K^{-1} = n^2 (\delta\mu/\delta n)_T$.

The third group of terms in $\mathcal{W}$ includes those in which $\tilde{c}_s$ appears in both the numerator and the denominator. Since

$$
\frac{\tilde{c}_s^2(\mathbf{q}^2)}{\omega^2 - \tilde{c}_s^2(\mathbf{q}^2)\mathbf{q}^2} = \frac{1}{|\mathbf{q}|^\alpha} \frac{c_s^2|\mathbf{q}|^\alpha + \omega_p^2}{\omega^2 - c_s^2\mathbf{q}^2 - \omega_p^2|\mathbf{q}|^{2-\alpha}} \sim
\begin{cases}
\frac{c_s^2 + \omega_p^2}{\omega^2} & (\alpha = 0) \\[2mm]
\frac{1}{|\mathbf{q}|^\alpha} \frac{\omega_p^2}{\omega^2} & (0 < \alpha < 2) \\[2mm]
\frac{1}{\mathbf{q}^2} \frac{\omega_p^2}{\omega^2 - \omega_p^2} & (\alpha = 2)
\end{cases}
\tag{C.14}
$$

as $\mathbf{q} \to 0$ at $\omega \neq 0$, response functions of this kind may diverge in this limit, for $0 < \alpha \leq 2$. On the other hand, as is clear from the right hand side of Eq. (C.14), response functions in this group are identical in the superfluid and superconductor at $\omega = 0$ and $\mathbf{q} \neq 0$. A basic example is given by the term $B^2$ included in $B^2$ in the second line of Eq. (C.11), which implies the London diamagnetic response,

$$
\rho_L(\omega,\mathbf{q}) = -\frac{n_0}{m} \frac{1}{|\mathbf{q}|^\alpha} \frac{c_s^2|\mathbf{q}|^\alpha + \omega_p^2}{\omega^2 - c_s^2\mathbf{q}^2 - \omega_p^2|\mathbf{q}|^{2-\alpha}}.
\tag{C.15}
$$

Another term in this group is the $h^2$ term in the third line of Eq. (C.11), which implies that the "dynamic compressibility" is given by

$$K_d^{-1}(\omega, \mathbf{q}) = n_0 m \frac{\omega^2}{|\mathbf{q}|^\alpha} \frac{c_s^2 |\mathbf{q}|^\alpha + \omega_p^2}{\omega^2 - c_s^2 \mathbf{q}^2 - \omega_p^2 |\mathbf{q}|^{2-\alpha}}. \tag{C.16}$$

The dynamic compressibility is defined as the pressure response to volume changes, $K_d^{-1} = -4\delta^2 \mathcal{W}/\delta h^2$, and vanishes at $\omega = 0$. On the other hand, $\lim_{\omega\to\infty} K_d^{-1}(\omega, \mathbf{q}) = K^{-1}(\mathbf{q})$, and $K_d^{-1}(\omega, \mathbf{q}) \sim K^{-1}(\mathbf{q}) \sim q^{-\alpha}$ as $q \to 0$.

The mixed responses $\kappa^{ij,k} = \delta T^{ij}/\delta E_k = 2\delta J^k/\delta \partial_t h_{ij}$ and $\chi^{ij} = 2\delta n/\delta h_{ij}$ are encoded in $E^i \partial_i h$ coming from $hq_i E^i$ in the third line of (C.11),

$$\kappa^{ij,k}(\omega, \mathbf{q}) = n_0 \delta^{ij} i q^k \frac{1}{|\mathbf{q}|^\alpha} \frac{c_s^2 |\mathbf{q}|^\alpha + \omega_p^2}{\omega^2 - c_s^2 \mathbf{q}^2 - |\mathbf{q}|^{2-\alpha}\omega_p^2}, \tag{C.17}$$

$$\chi^{ij}(\omega, \mathbf{q}) = -n_0 \delta^{ij} |\mathbf{q}|^{2-\alpha} \frac{c_s^2 |\mathbf{q}|^\alpha + \omega_p^2}{\omega^2 - c_s^2 \mathbf{q}^2 - |\mathbf{q}|^{2-\alpha}\omega_p^2}. \tag{C.18}$$

Finally, the second odd viscosity comes from $hq_i(\partial_t \omega^i - \partial^i \omega_t)$ contained in $hq_i E^i$ in the induced action, and is given by [Eq. (4.19)]

$$\eta_o^{(2)}(\omega, \mathbf{q}^2) = -\frac{1}{2} s n_0 \frac{1}{|\mathbf{q}|^\alpha} \frac{c_s^2 |\mathbf{q}|^\alpha + \omega_p^2}{\omega^2 - c_s^2 |\mathbf{q}|^2 - |\mathbf{q}|^{2-\alpha}\omega_p^2}. \tag{C.19}$$

# D Details of microscopic calculation

In this Appendix we provide details for the computations outlined in Section 5.

## D.1 Flat space induced action

In flat space ($h_{ij} = 0$), we rewrite the action (5.9) for the Nambu spinors in Fourier space as

$$S[\mathbf{\Psi}^\dagger, \mathbf{\Psi}, \mathcal{A}] = -\frac{1}{2} \int_q \mathbf{\Psi}_q^\dagger \mathcal{G}_{0,q}^{-1} \mathbf{\Psi}_q + \frac{1}{2} \int_{q,q'} \mathbf{\Psi}_q^\dagger [\Gamma_{1,q,-q'} + \Gamma_{2,q,-q'}] \mathbf{\Psi}_{q'}, \tag{D.1}$$

where the mean field propagator is given by Eq. (5.12) and we have split the vertex $\Gamma = \mathcal{G}_0^{-1} - \mathcal{G}^{-1}$ introduced in Eq. (5.16) into two vertices $\Gamma_{1,2}$ defined by

$$\Gamma_{1,q,-q'} = -i\mathcal{A}_{0,q-q'}\sigma_z - \frac{1}{2m}(\mathbf{q} + \mathbf{q}') \cdot \mathcal{A}_{q-q'}\sigma_0, \quad \Gamma_{2,q,-q'} = \frac{1}{2m} \int_p \mathcal{A}_p \cdot \mathcal{A}_{q-q'-p}\sigma_z. \tag{D.2}$$

In terms of these vertices, the contributions to Eq. (5.17) to first and second order in $\mathcal{A}$ are

$$S[\mathcal{A}] = \frac{1}{2} \text{Tr}[\mathcal{G}_0 \Gamma_1] + \frac{1}{2} \text{Tr}[\mathcal{G}_0 \Gamma_2] + \frac{1}{4} \text{Tr}[\mathcal{G}_0 \Gamma_1 \mathcal{G}_0 \Gamma_1]. \tag{D.3}$$

The first two sums yield

$$-\frac{1}{2} \text{Tr}\,\mathcal{G}_0 \Gamma_1 = -i\mathcal{A}_{0,q=0} n_0, \qquad -\frac{1}{2} \text{Tr}\,\mathcal{G}_0 \Gamma_2 = \frac{n_0}{2m} \int_p \mathcal{A}_p \cdot \mathcal{A}_{-p}, \tag{D.4}$$

with $n_0 = \frac{1}{2}\langle \Psi^\dagger \sigma_z \Psi \rangle = \frac{1}{2}\operatorname{Tr}[\mathcal{G}_0 \sigma_z]$ the density of electrons in the ground state in absence of external sources. The first term does not contribute to the response functions as it is linear in $A_\mu$ and $\partial_\mu \theta$ vanishes when evaluated at $q = 0$.

To evaluate the last integral, we use that with the shorthand notations $\kappa^\mu$ [Eq. (5.21)],

$$\Gamma_{1,p+q,-q} = -(\mathrm{i}\mathcal{A}_{0,p}\kappa^0_{p,q} + \mathcal{A}_{j,p}\kappa^j_{p,q}), \tag{D.5}$$

hence

$$
\begin{aligned}
\frac{1}{4}\operatorname{Tr}[\mathcal{G}_0\Gamma_1\mathcal{G}_0\Gamma_1] = \frac{1}{2}\int_q \Bigg\{ &\mathcal{A}_{i,-q}\mathcal{A}_{j,q}\int_p \frac{1}{2}\operatorname{tr}[\kappa^i_{p,q}\mathcal{G}_{0,p}\kappa^j_{p,q}\mathcal{G}_{0,p+q}] \\
&+ (\mathrm{i}\mathcal{A}_{0,-q})\mathcal{A}_{j,q}\int_p \frac{1}{2}\operatorname{tr}[\kappa^0_{p,q}\mathcal{G}_{0,p}\kappa^j_{p,q}\mathcal{G}_{0,p+q}] \\
&+ \mathcal{A}_{i,-q}(\mathrm{i}\mathcal{A}_{0,q})\int_p \frac{1}{2}\operatorname{tr}[\kappa^i_{p,q}\mathcal{G}_{0,p}\kappa^0_{p,q}\mathcal{G}_{0,p+q}] \\
&+ (\mathrm{i}\mathcal{A}_{-q,0})(\mathrm{i}\mathcal{A}_{q,0})\int_p \frac{1}{2}\operatorname{tr}[\kappa^0_{p,q}\mathcal{G}_{0,p}\kappa^0_{p,q}\mathcal{G}_{0,p+q}] \Bigg\}. 
\end{aligned}
\tag{D.6}
$$

Introducing the correlators $Q^{\mu\nu}(q)$ [Eq. (5.21)] and transforming back to real time gives Eq. (5.20). Integrating out the phase mode $\theta$ yields Eq. (5.22).

## D.2 Conductivity from the loop integrals

The correlations functions $Q^{\mu\nu}(q)$ are expressed in terms of the normal and anomalous propagators $G_q$ and $F_q$ ,

$$Q^{00}(q) = \int_p G_p G_{p+q} - \operatorname{Re}[F_p^* F_{p+q}], \tag{D.7}$$

$$Q^{i0}(q) = \frac{1}{2m}\int_p (q^i + 2p^i)\{G_p G_{p+q} + \mathrm{i}\operatorname{Im}[F_p^* F_{p+q}]\}, \tag{D.8}$$

$$Q^{ij}(q) = \frac{n_0}{m}\delta^{ij} + \frac{1}{4m}\int_p (q^i + 2p^i)(q^j + 2p^j)\{G_p G_{p+q} + \operatorname{Re}[F_p^* F_{p+q}]\}. \tag{D.9}$$

These expressions can be further simplified by exploiting the space rotation symmetry of the theory and noticing that the functions are either odd or even under $\omega_n \to -\omega_n$. In particular, $Q^{i0}(\mathrm{i}\omega_n, \mathbf{q})$ can be split into an even and an odd part,

$$Q^{i0}(\mathrm{i}\omega_n, \mathbf{q}) = Q^{i0}_{\mathrm{e}}(\mathrm{i}\omega_n, \mathbf{q}) + Q^{i\tau}_{\mathrm{o}}(\mathrm{i}\omega_n, \mathbf{q}), \tag{D.10}$$

$$Q^{i0}_{\mathrm{e}}(q) = \frac{1}{2m}\int_p (q^i + 2p^i)G_p G_{p+q}, \quad Q^{i0}_{\mathrm{o}}(q) = \frac{\mathrm{i}}{2m}\int_p (q^i + 2p^i)\operatorname{Im} F_p^* F_{p+q}. \tag{D.11}$$

While both $Q^{i0}_{\mathrm{e}}$ and $Q^{i0}_{\mathrm{o}}$ transform like vectors, the latter is odd under both time-reversal and parity. The decomposition for all $Q^{\mu\nu}(\mathrm{i}\omega_n, \mathbf{q})$ reads

$$Q^{00}(\mathrm{i}\omega_n, \mathbf{q}) = Q^\tau(\omega_n^2, \mathbf{q}^2), \tag{D.12}$$

$$Q^{i0}_{\mathrm{e}}(\mathrm{i}\omega_n, \mathbf{q}) = -\mathrm{i}\omega_n q^i Q^{\mathrm{e}}(\omega_n^2, \mathbf{q}^2), \quad Q^{i0}_{\mathrm{o}}(\mathrm{i}\omega_n, \mathbf{q}) = \mathrm{i}\epsilon^{ik}q_k Q^{\mathrm{o}}(\omega_n^2, \mathbf{q}^2), \tag{D.13}$$

$$Q^{ij}(\mathrm{i}\omega_n, \mathbf{q}) = \delta^{ij}Q^A(\omega_n^2, \mathbf{q}^2) + q^i q^j Q^B(\omega_n^2, \mathbf{q}^2). \tag{D.14}$$

Each of the five scalar functions $Q^\tau$, $Q^{\mathrm{e}}$, $Q^{\mathrm{o}}$, $Q^A$, $Q^B$ can be obtained by projecting the $Q^{\mu\nu}$ onto the corresponding tensors.

This decomposition allows us to write the projection of the conductivity tensor onto its antisymmetric part as

$$\sigma^H(\omega, \mathbf{q}) = \frac{\epsilon_{ij}}{2} \frac{1}{i\omega^+} K^{R,ij}(\omega, \mathbf{q}) = \frac{\mathbf{q}^2 Q^o(\mathbf{q}^2 Q^B + Q^A + \omega^2 Q^e)}{\mathbf{q}^4 Q^B + \mathbf{q}^2 Q^A + \omega^2(2\mathbf{q}^2 Q^e + Q^\tau)}, \tag{D.15}$$

with the functions on the rhs evaluated at the momentum $\mathbf{q}$ and analytically continued to the real frequency $\omega$. The polarisation bubbles are regular at small $\mathbf{q}$ and $\omega$ and in that limit Eq. (D.15) reduces to

$$\sigma^H(\omega \to 0, \mathbf{q} \to 0) = Q^o(0,0) \frac{\mathbf{q}^2 Q^A(0,0)}{\mathbf{q}^2 Q^A(0,0) + \omega^2 Q^\tau(0,0)} \tag{D.16}$$

up to order $\omega^2$, $\mathbf{q}^2$. These loop integrals evaluate to

$$Q^\tau(0,0) = -\frac{m}{2\pi}, \qquad Q^A(0,0) = \frac{n_0}{m}, \qquad Q^o(0,0) = \pm\frac{1}{8\pi}. \tag{D.17}$$

Substituting these results into Eq. (D.16) gives rise to Eq. (5.26).

### D.3 Pure geometric induced action

In this Appendix, we determine the contributions to the odd viscosities coming from $S_0[h]$. In the first step we compute $S_0[h]$ to quadratic order in $h_{ij} = g_{ij} - \delta_{ij}$. To that end, we expand $\Gamma = \mathcal{G}_0^{-1} - \mathcal{G}^{-1}$, defined by Eq. (5.16), in powers of $h$, setting $\mathcal{A} = 0$.

First, we recall the expansions

$$\sqrt{g} = 1 + \frac{1}{2}\delta^{ij}h_{ij} + \frac{1}{2}h - \frac{1}{8}\delta^{ij}\delta^{kl}h_{ij}h_{kl} + \mathcal{O}(h^3), \tag{D.18}$$

$$e^{ia} = \delta^{ia} - \frac{1}{2}h^{ia} + \frac{3}{8}h^{ik}h_k{}^a + \mathcal{O}(h^3). \tag{D.19}$$

In the above expressions, we denote $h = \det(h_{ij}) = \epsilon^{ik}\epsilon^{jl}h_{ij}h_{kl}$ and the indices of $h_{ij}$ (and $\delta_{ij}$) are raised and lowered using the flat metric, as well as the second index of the vielbeins. Using this, one has

$$\begin{aligned}
\Gamma_{x,x'} = &-\left(\frac{1}{2}\delta^{ij}h_{ij} + \frac{1}{2}h - \frac{1}{8}\delta^{ij}\delta^{kl}h_{ij}h_{kl}\right)\mathcal{G}_0^{-1} - \left(1 + \frac{1}{2}\delta^{ij}h_{ij}\right) \\
&\times \left\{\frac{1}{4}\partial_\tau\left(\delta^{ij}h_{ij} + h - \frac{1}{2}\delta^{ij}\delta^{kl}h_{ij}h_{kl}\right)\right. \\
&\left. - \frac{h_{ij}p^i p^j}{2m} + \Delta(\frac{1}{2}h_{ia} - \frac{1}{8}h_{ik}h^k{}_a)p^i\tilde{\sigma}^a\right\}\delta(x-x') + \mathcal{O}(h^3).
\end{aligned} \tag{D.20}$$

There are two contributions of quadratic order in $h_{ij}$ to the induced action: $\mathrm{Tr}\,\Gamma^{(2)}\mathcal{G}_0$ and $\mathrm{Tr}\,\Gamma^{(1)}\mathcal{G}_0\Gamma^{(1)}\mathcal{G}_0$, with $\Gamma^{(i)}$ the term of $i$-th order in $h_{ij}$ in $\Gamma$. A first remark is that the tadpole term $\mathrm{Tr}\,\Gamma^{(2)}\mathcal{G}_0$ does not contribute to the anti-symmetric part of the viscosity. To see it, one notes that $\Gamma^{(2)}$ can be expressed as

$$\Gamma^{(2)}_{x,x'} = [h_{ij}h_{kl}D_1^{ij,kl} + \partial_\tau(h_{ij}h_{kl})D_2^{ij,kl}]\delta(x-x'), \tag{D.21}$$

with $D_{1,2}^{ij,kl}$ differential operators satisfying $D_{1,2}^{ij,kl} = D_{1,2}^{kl,ij}$. The corresponding induced action is $\propto \sum_p h_{-p,ij}h_{p,kl}\int_q \mathcal{G}_0(q)D_1^{ij,kl}(q)$ which is symmetric under the exchange of the two pairs of indices $(ij)$ and $(kl)$.

It is thus sufficient to expand $\Gamma$ to order one in $h_{ij}$ and, in Fourier space,

$$
\begin{aligned}
\Gamma^{(1)}_{q,-q'} = h_{ij}(q-q')\Bigg\{ &\frac{1}{2}\delta^{ij}\left(q'_0\sigma^0 - \frac{q_k q'^l}{2m}\sigma^z + \Delta q'_k \tilde{\sigma}^k\right) \\
&+ \left[\frac{1}{4}(q-q')_0\delta^{ij}\sigma^0 - \frac{q^i q'^j + q'^i q^j}{4m}\sigma^z + \frac{1}{4}\Delta[q'^i \tilde{\sigma}^j + \tilde{\sigma}^i q'^j]\right]\Bigg\},
\end{aligned}
\tag{D.22}
$$

which yields the induced action $S_0 = \text{Tr}[\Gamma^{(1)}\mathcal{G}_0\Gamma^{(1)}\mathcal{G}_0]/4$, see (5.30).

## D.4 Pure geometric contribution to the odd viscosities

The odd viscosities are obtained by projecting $R^{ij,kl}$ onto the odd tensors $\sigma^{ab}$, using the identity $(\sigma^{ab})^{ijkl}(\sigma^{cd})^{jilk}/8 = \delta_{a,c}\delta_{b,d} - \delta_{a,d}\delta_{b,c}$. All loop integrals obtained while computing the trace are regular at small $\mathbf{q}$, $\omega$ and the contribution to $\eta_o^{(2)}$ is of order $\mathcal{O}(|\mathbf{q}|,\omega)$. As for $\eta_o^{(1)}$, one has in terms of the propagators (5.14) and (5.15)

$$
\begin{aligned}
\eta_o^{(1)}(\omega,\mathbf{q}) = \frac{1}{i\omega}\frac{is\Delta}{16m}\int_p \mathbf{p}(\mathbf{p}+\mathbf{q})\{&[\mathbf{p}(2\mathbf{p}+\mathbf{q})f_p + m\Delta G_p]G_{-p-q} \\
&- [\mathbf{p}(2\mathbf{p}+\mathbf{q})f_p + m\Delta G_{-p}]G_{p+q}\}.
\end{aligned}
\tag{D.23}
$$

To compute this expressions, we first recall that the fermion density is given by

$$
\begin{aligned}
n_0 &= \frac{1}{2}\text{Tr}[\mathcal{G}_0\sigma_z] = -\frac{1}{2}\int_{\mathbf{q},\omega_n}\frac{(i\omega_n + \xi_\mathbf{q})e^{i\omega_n \nu} + (-i\omega_n + \xi_\mathbf{q})e^{-i\omega_n \nu}}{\omega_n^2 + \mathbf{q}^2\Delta^2 + \xi_\mathbf{q}^2} \\
&= \frac{1}{2}\int_\mathbf{q} 1 - \frac{\xi_\mathbf{q}}{\sqrt{\mathbf{q}^2\Delta^2 + \xi_\mathbf{q}^2}} = \int_\mathbf{q}\frac{\mathbf{q}^2\Delta^2(\mathbf{q}^2/m - \xi_\mathbf{q})}{4(\mathbf{q}^2\Delta^2 + \xi_\mathbf{q}^2)^{3/2}}.
\end{aligned}
\tag{D.24}
$$

We have introduced a convergence factor $\nu \to 0^+$ to take care of the time ordering of the operators. The last integral is obtained by performing an integration by parts $\sum_\mathbf{q} f(\mathbf{q}^2) = -\sum_\mathbf{q}\mathbf{q}^2 f'(\mathbf{q}^2)$, discarding the boundary term. Evaluating Eq. (D.23) at $q = 0$ and carrying the integral over Matsubara frequencies, one gets

$$
\eta_o^{(1)} = s\int_\mathbf{q}\frac{\mathbf{q}^2\Delta^2(\mathbf{q}^2/m - \xi_\mathbf{q})}{8(\mathbf{q}^2\Delta^2 + \xi_\mathbf{q}^2)^{3/2}} = \frac{sn_0}{2},
\tag{D.25}
$$

which is Eq. (5.34) in the main text.

## D.5 Phase term contribution to the odd viscosities

In this Appendix, we determine the contribution to the odd viscosities from the phase mode part of the induced action. To that end, we determine first $N^\mu[h]$, the coefficient of the term linear in $\mathcal{A}_\mu$ in the induced action $S[\mathcal{A},h]$. We expand $\Gamma$, defined by Eq. (5.16) in powers of $\mathcal{A}$.[17] Denoting $\Gamma^{(i)}$ the term of order $\mathcal{O}(\mathcal{A}^i)$,[18] we have

$$
\Gamma^{(0)}_{p,-p'} = \Delta_0 p'_i(e^{ia}_{p-p'} - \delta^{ia}\delta_{p,p'})\tilde{\sigma}_a + \frac{p_i p'_j + p'_i p_j}{4m}(g^{ij}_{p-p'} - \delta^{ij}\delta_{p,p'})\sigma_z,
\tag{D.26}
$$

$$
\Gamma^{(1)}_{p,-p'} = \int_q \mathcal{A}_{-q,\mu}\gamma^{(1)\mu}_{p,-p',q}, \quad \gamma^{(1)0}_{p,-p',q} = -e\delta_{p-p'+q}\sigma^z, \quad \gamma^{(1)i}_{p,-p',q} = -\frac{e}{2m}(p_j + p'_j)g^{ij}_{p-p'+q}\sigma^0.
\tag{D.27}
$$

---

[17]Anticipating the analytic continuation we replace $i\mathcal{A}_\tau$ by $\mathcal{A}_\tau$.

[18]Note that this convention is different to that of Appendix D.3 where $\mathcal{A}$ is dropped and $\Gamma^{(i)}$ denotes the terms of order $\mathcal{O}(h^i)$.

By identifying the induced action (5.17) with the expansion (5.19), we find

$$N_q^\mu[h] = \frac{1}{2} \operatorname{tr} \int_p \mathcal{G}_{0,p} \gamma_{p,-p,q}^{(1)\mu} + \frac{1}{2} \int_p \operatorname{tr} \mathcal{G}_{0,p} \gamma_{p,-p-q,q}^{(1)} \mathcal{G}_{0,p+q} \Gamma_{p+q,-p}^{(0)}. \tag{D.28}$$

Computing $N_q^\mu$ at order $\mathcal{O}(h)$ is done by evaluating the bubble diagram $\operatorname{Tr} \mathcal{G}_0 \Gamma^{(1)} \mathcal{G}_0 \Gamma^{(0)}$, setting $h_{ij} = 0$ in $\Gamma^{(1)}$ and keeping $\Gamma^{(0)}$ to linear order in $h_{ij}$, which produces Eqs. (5.35) and (5.36).

To project the viscosity tensor deduced from Eqs. (5.37) and (5.38) onto the antisymmetric tensors $\sigma^{ab}$, we expand $I_q^{ij}$ in the limit of small momentum at order $\mathcal{O}(\mathbf{q}^2)$. $I_q^{ij}$ can be decomposed onto symmetric rank-two tensors that transform like $\mathbf{q}$ under SO(2) rotations,[19]

$$I_q^{ij} = [I_\delta(\mathrm{i}\omega_n) + \mathbf{q}^2 I_{\delta,1}(\mathrm{i}\omega_n)]\delta^{ij} + [I_{\mathbf{uu}}(\mathrm{i}\omega_n) + \mathbf{q}^2 I_{\mathbf{uu},1}(\mathrm{i}\omega_n)]u^{(i} u^{*j)}$$
$$+ I_{\mathbf{qu}}(\mathrm{i}\omega_n)q^{(i} u^{j)}(\mathbf{q} \cdot \mathbf{u}^*) + I_{\mathbf{qu}^*}(\mathrm{i}\omega_n)q^{(i} u^{*j)}(\mathbf{q} \cdot \mathbf{u}^*) + I_{\mathbf{qq}}(\mathrm{i}\omega_n)q^i q^j + \mathcal{O}(\mathbf{q}^2), \tag{D.29}$$

where the integrals $I_\alpha$ are functions of the external frequency $\mathrm{i}\omega_n$.

No matter what are the values of the integrals $I_\alpha$, the projection of $S^{ijkl}$ onto $\sigma^{xz}$ vanishes and there is no contribution to $\eta_\mathrm{o}^{(1)}$. We now extract the viscosity $\eta_\mathrm{o}^{(2)}$ from the integrals $I_\alpha$ defined by Eq. (D.29). Expanding at small frequencies and noting $I_\alpha(\mathrm{i}\omega_n) = I_\alpha(0) + \mathrm{i}\omega_n I_\alpha'(0) + \mathcal{O}(\omega_n^2)$, the projection yields

$$\eta_\mathrm{o}^{(2)}(\mathbf{q},\omega) = 8s \frac{I_{\mathbf{qu}}(0)[I_\delta'(0) + I_{\mathbf{uu}}'(0)]}{n/m(c_s^2\omega^2 - \mathbf{q}^2)}. \tag{D.30}$$

To get this result, we have used that $I_{\mathbf{qu}}(0) = -I_{\mathbf{qu}^*}(0)$, $I_{\mathbf{qu}}'(0) = I_{\mathbf{qu}^*}'(0)$, and as shown in Appendix D.2 $q_\alpha q_\beta Q^{\alpha\beta}(q) = -n/m(c_s^{-2}\omega^2 - \mathbf{q}^2)$ at small frequency and momentum. The relevant integrals are

$$I_\delta(\mathrm{i}\omega_n) = -\frac{\mathrm{i}\omega_n}{8m} \int_{\mathbf{p},\mathrm{i}\omega_m} \mathbf{p}^2 [\mathbf{p}^2 f(\mathbf{p},\mathrm{i}\omega_m)f(\mathbf{p},\mathrm{i}\omega_m) - G(-\mathbf{p},-\mathrm{i}\omega_m)G(-\mathbf{p},-\mathrm{i}\omega_m)$$
$$- G(\mathbf{p},\mathrm{i}\omega_m)G(\mathbf{p},\mathrm{i}\omega_m)] + \mathcal{O}(\omega_n^2), \tag{D.31}$$

$$I_{\mathbf{uu}}(\mathrm{i}\omega_n) = \frac{\mathrm{i}\omega_n}{8m} \int_{\mathbf{p},\mathrm{i}\omega_m} \mathbf{p}^2 \{f(\mathbf{p},\mathrm{i}\omega_m)[2\Delta m(G(-\mathbf{p},-\mathrm{i}\omega_m) + G(\mathbf{p},\mathrm{i}\omega_m)) - \mathbf{p}^2 f(\mathbf{p},\mathrm{i}\omega_m)]$$
$$+ 2\Delta m(G(-\mathbf{p},-\mathrm{i}\omega_m) + G(\mathbf{p},\mathrm{i}\omega_m))f(\mathbf{p},\mathrm{i}\omega_m)\} + \mathcal{O}(\omega_n^2), \tag{D.32}$$

$$I_{\mathbf{qu}}(\mathrm{i}\omega_n) = \frac{4\Delta m}{32m^2} \int_{\mathbf{p},\mathrm{i}\omega_m} \mathbf{p}^2 G(-\mathbf{p},-\mathrm{i}\omega_m)[f(\mathbf{p},\mathrm{i}\omega_m) + \mathbf{p}^2 f'(\mathbf{p},\mathrm{i}\omega_m)]$$
$$- |\mathbf{p}|^4 f(\mathbf{p},\mathrm{i}\omega_m)[f(\mathbf{p},\mathrm{i}\omega_m) + 4\Delta m G'(\mathbf{p},\mathrm{i}\omega_m)] + \mathcal{O}(\omega_n), \tag{D.33}$$

where $G' = \partial_{\mathbf{q}^2} G$, $f' = \partial_{\mathbf{q}^2} f$. As in Appendix D.3, we perform for each integral the sum over Matsubara frequencies and obtain ultraviolet-divergent integrals. We regularize them using a hard momentum cutoff and compare these integrals to those defining the density by identifying the leading logarithmic term. We thus obtain $I_{\mathbf{qu}}(0) = n_0/32m$, $I_\delta'(0) + I_{\mathbf{uu}}'(0) = -2n_0$ to finally get Eq. (5.39).

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
