# Peer review of "Hall viscosity and conductivity of two-dimensional chiral superconductors"

_SciPost Physics, doi:SciPost Phys. 9, 006 (2020)_

## Round 1 · Referee Report · Anonymous (Referee 1) · 2020-5-14

Strengths

1 - Clarity and consistency of notations
2 - New results for Hall conductivity and Hall viscosity for chiral superconductors with instantaneous interaction
3 - Nice notations allowing comparison between the results for superfluids and superconductors
4 - Details of computations are presented both in the main text and in very useful appendices

Weaknesses

No particular weaknesses

Report

The manuscript presents the computation of Hall conductivity and two Hall viscosities for chiral superfluids and superconductors with Galilean symmetry. The interaction in superconductors is taken into account in the instantaneous approximation. The main physical result is the qualitative difference between superconductors and superfluids in the case when interparticle interaction is long-ranged. In particular, for the intrinsic superconductor (1/q^2 interaction) the second Hall viscosity has a 1/q^2 singularity and modifies Hall viscosity even in the limit q\to 0. For superconductors with unscreened interaction through the internal gauge field this effect might be observable, in principle.

The authors analyze the symmetries of the problem in section 3 and derive Galilean Ward identities. Then they check these identities for every computation they do in the rest of the manuscript. The computation is done using the effective field theory approach in section 5. The main results are obtained with phenomenological coefficients. Then it is repeated starting from BCS microscopic model and the values of coefficients are obtained in terms of microscopic parameters.

While the computation is conceptually simple it requires a lot of careful bookkeeping. The authors did a very good job at it and even managed to present all necessary details in a readable form. I found the appendices giving some missing details very useful.

To conclude, I believe, that the presented results are valid and well presented. Therefore, the manuscript deserves publication in SciPost. It can be published as-is. I give few very minor comments on the text below.

Requested changes

1 - On page 6 the authors refer to (2.4) as "exotic dissipationless response..." This is true. In hydro literature this term is known as the term "breaking objectivity". Objectivity is the property of the symmetry of viscosity tensor with respect to last two indices.

2 - On page 6, it seems that the quantity $u$ used in the last equation on the page is not introduced prior to that equation. One can guess that this is a trace of the strain $u=u_{ii}$ but it would be nice to have this explained somewhere.

3 - In eq. 3.9 it would be nice to use round brackets around $q^q^j/q^2$ to separate it from $\sigma_L$.

4 - On page 20 between eqs 5.26 and 5.27 the sentence "Let us now discuss what makes here possible the appearance of a finite Hall conductivity." does not sound right.

  • validity: top
  • significance: high
  • originality: high
  • clarity: top
  • formatting: perfect
  • grammar: perfect

Author:  Félix Rose  on 2020-06-16  [id 855]

(in reply to Report 1 on 2020-05-14)
Category:
answer to question
correction

We are grateful to the referee for carefully reading our paper and providing valuable feedback.
Below are our point-by-point replies and changes we made based on the comments of the referee.

1 - “On page 6 the authors refer to (2.4) as "exotic dissipationless response..." This is true. In hydro literature this term is known as the term "breaking objectivity". Objectivity is the property of the symmetry of viscosity tensor with respect to last two indices.”

The viscosity tensor we compute does satisfy the symmetry properties $\eta^{ijkl}=\eta^{jikl}=\eta^{ijlk}$ by construction, as it is related to functional derivatives of the free energy wrt a spacetime dependent metric $g_{ij}$ satisfying $g_{ij}=g_{ji}$.
The exotic response rather comes from the momentum dependence of the odd viscosity, in turn a consequence of time-reversal and parity breaking.

To emphasize this, we added to the manuscript a paragraph between Eqns. (2.4) and (2.5), as well as the footnote 3.

2 - “On page 6, it seems that the quantity u used in the last equation on the page is not introduced prior to that equation. One can guess that this is a trace of the strain $u=u_{ii}$ but it would be nice to have this explained somewhere.”

$u=u_{ii}=\nabla \dot \xi$ is defined in the paragraph below (2.2).

3 - “In eq. 3.9 it would be nice to use round brackets around $q^i q^j/q^2$ to separate it from $σ_L$.”

The modification has been implemented.

4 - “On page 20 between eqs 5.26 and 5.27 the sentence "Let us now discuss what makes here possible the appearance of a finite Hall conductivity." does not sound right.”

We replaced the sentence by “In the above calculation, two ingredients are necessary to obtain a non-vanishing Hall conductivity.”

---

## Round 1 · Referee Report · Anonymous (Referee 2) · 2020-5-24

Strengths

  1. Clarity of presentation
  2. Several consistency checks of results
  3. Surprising singular feature in response functions

Weaknesses

None

Report

This paper studies non-dissipative transport in chiral superconductors, with a focus on parity-odd transport. The problem is addressed by taking advantage of an existing effective field theory (EFT) describing the low energy physics of chiral superfluids, and adding an instantaneous Coulomb interaction. The authors then study linear response by expanding the EFT up to quadratic order in fields in the presence of background sources, and performing the Gaussian integral to obtain the generating functions. This produces response functions in terms of a few EFT parameters, which can be computed given a microscopic model; such a model is studied in Sec.5. The authors derive Ward identities that elegantly constrain transport, and verify that they are satisfied in the EFT. These calculations reveal a peculiar 1/q^2 singularity in response functions in these systems.

The paper is very clearly written, with motivation well established and the results pass a number of consistency checks made by the authors. I therefore recommend publication, after a few minor points are addressed (see below).

Requested changes

  1. The Gaussian approximation to the EFT is possible because interactions are irrelevant. These will lead to higher loop corrections to transport, which are suppressed at small wavevector and frequency. Higher derivative terms will lead to similar corrections. It would be useful for the reader if an estimate of these corrections could be made (which power of q and \omega?)

  2. Energy transport is not mentioned. Is it also tied to charge transport by Galilean invariance?

  3. Below Eq. (3.9), the authors write "[..] while the Hall component fixes its antisymmetric part [..] and describes dissipationless transport of particles.". This statement applies to the dc limit. However the authors are interested in finite frequency transport, where the Hall component can also contain a dissipative part (e.g. a linear in frequency contribution to \sigma_H). The same comment holds for the viscosity, below (3.10).

Typos: symeterization, Majarona

  • validity: top
  • significance: high
  • originality: high
  • clarity: top
  • formatting: perfect
  • grammar: perfect

Author:  Félix Rose  on 2020-06-16  [id 856]

(in reply to Report 2 on 2020-05-24)
Category:
answer to question
correction
pointer to related literature

We are grateful to the referee for carefully reading our paper and providing valuable feedback.
Below are our point-by-point replies and changes we made based on the comments of the referee.

1 - “The Gaussian approximation to the EFT is possible because interactions are irrelevant. These will lead to higher loop corrections to transport, which are suppressed at small wavevector and frequency. Higher derivative terms will lead to similar corrections. It would be useful for the reader if an estimate of these corrections could be made (which power of q and \omega?)”

The order of magnitude of the next-to-leading corrections to the odd conductivity and viscosity are derived in Appendix C.1.3, see below (C.10).
We refer to it in the main text in the paragraph above (4.7).

2 - “Energy transport is not mentioned. Is it also tied to charge transport by Galilean invariance?”

To the extend of our knowledge, Galilean invariance alone is not enough to relate in a simple manner charge and energy transport.
This is an interesting question, which goes beyond the scope of this paper.
If one is interested in investigating it, the formalism to compute energy transport coefficients with sources coupling to energy currents is developed in Phys. Rev. D, 2015, 91, 045030.
For two-dimensional p+ip superfluids, this formalism was used to compute the energy density and current in Phys. Rev. B 91, 064508 (2015).

3 - “Below Eq. (3.9), the authors write "[..] while the Hall component fixes its antisymmetric part [..] and describes dissipationless transport of particles.". This statement applies to the dc limit. However the authors are interested in finite frequency transport, where the Hall component can also contain a dissipative part (e.g. a linear in frequency contribution to \sigma_H). The same comment holds for the viscosity, below (3.10).”

It is true that odd powers in omega of the Hall conductivity and odd viscosity would lead to dissipative transport. However, for a system at equilibrium (described by a partition function), which is the case we consider, such odd powers are forbidden.

We explain this in the footnote 6.

“Typos: symeterization, Majarona”

The typos have been corrected.

---

## Editorial Decision

published